# History-Aware Transformation of ReID Features for Multiple Object Tracking

## Abstract

In Multiple Object Tracking (MOT), Re-identification (ReID) features are widely employed as a powerful cue for object association. However, they are often wielded as a one-size-fits-all hammer, applied uniformly across all videos through simple similarity metrics. We argue that this overlooks a fundamental truth: MOT is not a general retrieval problem, but a context-specific task of discriminating targets within a single video. To this end, we advocate for the adjustment of visual features based on the context specific to each video sequence for better adaptation. In this paper, we propose a history-aware feature transformation method that dynamically crafts a more discriminative subspace tailored to each video's unique sample distribution. Specifically, we treat the historical features of established trajectories as context and employ a tailored Fisher Linear Discriminant (FLD) to project the raw ReID features into a sequence-specific representation space. Extensive experiments demonstrate that our training-free method dramatically enhances the discriminative power of features from diverse ReID backbones, resulting in marked and consistent gains in tracking accuracy. Our findings provide compelling evidence that MOT inherently favors context-specific representation over the direct application of generic ReID features. We hope our work inspires the community to move beyond the naive application of ReID features and towards a deeper exploration of their purposeful customization for MOT. Our code will be released.

## 1 Introduction

Multiple Object Tracking (MOT) is a fundamental computer vision task that aims to detect objects and maintain their identities across video frames. Its primary goal is to generate a distinct trajectory for each target by associating its corresponding detections over time. As a critical component for understanding dynamic scenes, MOT serves as an essential prerequisite for a wide range of downstream applications, such as autonomous driving, human behavior analysis, trajectory forecasting, and public surveillance.

The tracking-by-detection paradigm (Bewley et al., 2016; Zhang et al., 2022a; Cao et al., 2022) has long been the dominant and most widely adopted approach in the field of multiple object tracking. According to the task definition, it decouples the complex tracking problem into two sequential subtasks: first, an object detector localizes all targets within each frame, and second, an association algorithm links these detections across frames to form individual trajectories. As the former step is well-addressed by powerful detectors (Ge et al., 2021; Varghese & Sambath, 2024), the crux of this paradigm lies in the association stage. To solve this association problem, most methods (Zhang et al., 2021; Cao et al., 2022; Dendorfer et al., 2022) model existing trajectories with discriminative cues and then allocate identities by minimizing the matching cost.

Given that distinct targets often exhibit unique visual characteristics, appearance has emerged as a powerful and prevalent discriminative feature for trajectory modeling. In practice, visual features are typically extracted using off-the-shelf Re-Identification (ReID) models (Luo et al., 2019), and a cost matrix is then formulated by the cosine distance. Despite its demonstrated success (Wojke et al., 2017; Zhang et al., 2021; Aharon et al., 2022; Yang et al., 2023b), a latent contradiction persists within this paradigm. According to the definition, the goal of a general ReID model is to learn a universal feature representation capable of distinguishing any given identity from a large,

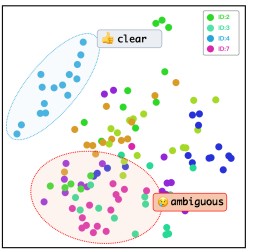 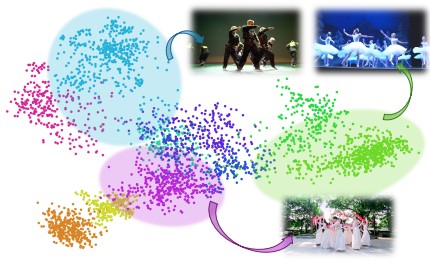

(a) High similarity between the ReID features of different trajectories can create potential association ambiguity.

(b) ReID features of individual sequences form compact clusters in the original space.

Figure 1: Visualization of ReID features in the original representation space (Luo et al., 2019).

open set. In contrast, the challenge within MOT is to discriminate only between the limited set of targets appearing in a specific video, which constitutes a more nuanced, expert-level requirement. As illustrated in Figure 1a, targets within the same video sequence always exhibit a high degree of similarity, making them difficult to distinguish in a generic, globally-trained feature space (Luo et al., 2019). Furthermore, this intra-sequence similarity causes their representations to cluster within a confined subspace of the original space, leading to redundancy, as shown in Figure 1b. Based on the foregoing observations, a natural question arises: *can we seek a specialized representation subspace for the MOT task, one that is more focused on distinguishing identities within the constrained set of a given sequence?*

In this paper, we first confirm the significant influence of representation discriminability on tracking performance. Accordingly, we posit that an ideal representation space should be structured for each sequence to minimize intra-trajectory distances while maximizing inter-trajectory distances. Such a configuration would effectively increase the separability between positive and negative pairs, thereby enhancing the tracking ability. Coincidentally, this principle is conceptually analogous to the objective of Fisher Linear Discriminant (FLD) (Fisher, 1938), where trajectories are viewed as different categories. Since tracking is an online, continuous inference process, the established historical context can be regarded as a dynamic approximation of the data distribution for the current sequence. Building on these discussions, in practice, we feed the existing trajectories as conditional input into the FLD algorithm. By computing the closed-form solution, we derive a projection matrix that maps the original features onto a more discriminative subspace tailored for separating different trajectories. The experiment results reveal that this simple yet effective feature transformation substantially enhances discriminability and boosts overall tracking performance. Nevertheless, we revisit this process and argue that MOT possesses some task-specific demands that should not be overlooked. Firstly, since a target's features gradually change over time in online tracking, we use a temporally weighted average to construct the trajectory's center, rather than the naïve averaging. This makes the resulting representation better suited for the similarity assessment required at the current timestep. Secondly, historical trajectories are not always reliable due to occlusions and tracking errors. Moreover, the tracker need to handle newborn objects, which are not considered in the transformed space. These challenges underscore the importance of retaining the original feature space with its strong robustness and generalization capabilities. To this end, we combine the similarity scores from both the general and specialized representations, thereby leveraging the complementary strengths of each.

To clearly validate the impact of our ReID feature transformation, our experiments are primarily conducted on trackers that rely solely on appearance (Luo et al., 2019; Li et al., 2024a). This approach minimizes the complex designs and potential interference introduced by other tracking cues (Welch et al., 1995; Bewley et al., 2016). In practice, we build a ReID-based tracker upon the most widely-used ReID model (Luo et al., 2019) in the MOT community (Yang et al., 2023b; Lv et al., 2024) and validate the effectiveness of our components. Relying solely on the ReID cue, our method achieves significant performance improvements. Remarkably, in some scenarios (Cui et al., 2023), our algorithm substantially outperforms methods that combine multiple clues (Aharon et al., 2022; Cui et al., 2023; Lv et al., 2024), establishing a new state-of-the-art result. This finding strongly indicates that the full potential of appearance information has been underestimated in past

research. We also confirm the generalization capability of our proposed method by applying it to Li et al. (2024a) with various visual encoders (He et al., 2016; Zhou et al., 2022; Kirillov et al., 2023; Liu et al., 2024), observing stable performance boosts across every case. Additionally, we conduct experiments on several hybrid-based methods (Cao et al., 2022; Yang et al., 2023b). The results demonstrate that our approach can be seamlessly integrated into these advanced trackers, achieving state-of-the-art performance.

To sum up, our main contribution include:

- Following our analysis in Section 2.2, we equip Fisher Linear Discriminant with historical tracklet supervision to transform ReID features, enhancing their discriminability.
- To address the practical needs of MOT task, we propose two customized components, *temporally-weighted trajectory centroid* (Section 3.2) and *knowledge integration* (Section 3.3), which further improve our tracking performance.
- To prove the effectiveness of our method, we conduct extensive experiments on ReID-based methods, demonstrating consistent performance gains across diverse scenarios (Table 1, 2 and 3). We also validate its versatility by seamlessly integrating it into hybird-based trackers, pushing their state-of-the-art performance even further.

## 2 PRELIMINARY

### 2.1 REID-BASED TRACKER

The tracking-by-detection paradigm (Bewley et al., 2016; Zhang et al., 2022a; Cao et al., 2022) treats multiple object tracking as a two-step process. First, an object detector $\mathcal{D}$ is employed to localize all targets in a given frame $I_t$. Subsequently, these detections are associated with established trajectories based on a cost matrix or used to initialize new tracks. Following our discussion in Section 1, we simplify our experimental scope by concentrating on trackers that use only appearance cues for data association. Given an object bounding box, $\boldsymbol{b}_{t,i}$, in the $t$-th frame, a feature extraction network $\Phi$ is applied to obtain the corresponding visual feature $\boldsymbol{f}_{t,i}$, often referred to as a re-identification (ReID) feature. It is used to represent the appearance of each detection and to construct the feature of each trajectory. In practice, while numerous methods (Wojke et al., 2017; Maggiolino et al., 2023; Yang et al., 2023b) for trajectory modeling exist, we adopt the widely-used Exponential Moving Average (EMA) update strategy due to its proven efficiency and effectiveness, as formulated below:

$$\hat{\boldsymbol{f}}_{t,\tau_j} = \lambda \boldsymbol{f}_{t,\tau_j} + (1-\lambda)\hat{\boldsymbol{f}}_{t-1,\tau_j}, \tag{1}$$

where $\hat{\boldsymbol{f}}_{t-1,\tau_j}$ represents the appearance feature of track $\tau_j$ aggregated up to timestep $t-1$, $\boldsymbol{f}_{t,\tau_j}$ is the ReID feature obtained from the extractor $\Phi$ at the current frame $I_t$, and $\lambda$ is a momentum coefficient, typically set to a small value close to 0, that controls the update ratio.

Once the aforementioned features are prepared, we compute the matching cost for each detection-trajectory pair using a similarity metric. A common practice is to use the cosine similarity, which is calculated as follows:

$$\text{Cost}(t, i, \tau_j) = 1 - \text{Sim}(t, i, \tau_j) = 1 - \frac{\boldsymbol{f}_{t,i} \cdot \hat{\boldsymbol{f}}_{t-1,\tau_j}}{\|\boldsymbol{f}_{t,i}\|\|\hat{\boldsymbol{f}}_{t-1,\tau_j}\|}. \tag{2}$$

Accordingly, a cost matrix is constructed for the current frame based on all potential assignments. The Hungarian algorithm is then employed to find the globally optimal matching solution. Following this, the features of the matched tracks are updated according to Equation 1, in preparation for the next time step.

### 2.2 DISCRIMINATIVE CAPABILITY ANALYSIS

As stated in Section 2.1, since the tracker relies solely on appearance features for discrimination, it is intuitive to assume that the discriminative power of the ReID features is directly correlated with the tracking performance.

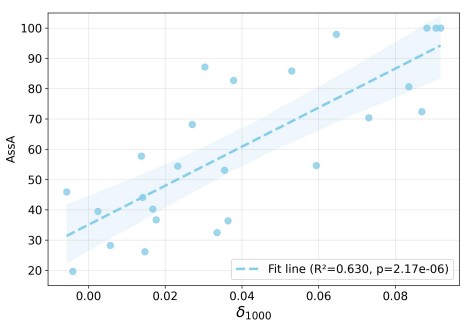 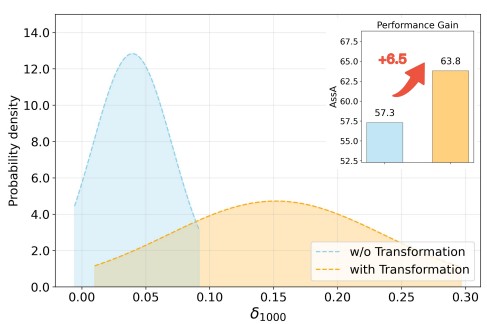

(a) Significant and reliable positive correlation between discriminability and tracking performance.

(b) Our transformation improves performance by enhancing feature discriminative capability.

Figure 2: Correlation between ReID feature discriminability $\delta_{1000}$ and tracking accuracy AssA on DanceTrack (Sun et al., 2022). Similar analysis on Cui et al. (2023) can be found in Figure 5.

To validate this hypothesis, it is necessary to quantify the discriminative capability of the representation space. Since the tracking process relies on cosine similarity for affinity measurement, as shown in Equation 2, we also adopt it as the cornerstone for evaluating the discriminability. To be more specific, we measure the discriminative power for the $i$-th detection in frame $t$ using a score, $\delta(t, i)$. This score is defined as the similarity margin between the detection's positive track and its most confusing negative track. Furthermore, since tracking failures are minority events within a given sequence, we focus on the most challenging cases. Therefore, for each video, we select the $1000$ worst scores and compute their average. This metric, termed $\delta_{1000}$, is used to quantify the discriminative ability of the ReID representations for a specific sequence (details in Section B.1). Accordingly, we conduct a statistical analysis on the representative dataset DanceTrack (Sun et al., 2022), as shown in Figure 2a. The results reveal a significant and reliable positive correlation between the discriminative capability ($\delta_{1000}$) of the ReID features and the object association accuracy (AssA (Luiten et al., 2021)). This conclusion provides a clear motivation for our work: to boost tracking performance by explicitly enhancing the discriminability of the representation space, described in Section 3.

## 2.3 FISHER LINEAR DISCRIMINANT

Fisher Linear Discriminant (FLD) (Fisher, 1938), also widely known as Linear Discriminant Analysis (LDA), is a classic supervised method used for both dimensionality reduction and classification. The core principle is to find a linear transformation that projects high-dimensional data onto a lower-dimensional space where the classes are maximally separated. In other words, the projection pulls the means of different classes far apart while keeping the data within each class tightly clustered. Mathematically, this is achieved by defining a within-class scatter matrix, $\boldsymbol{S}_W$, and a between-class scatter matrix, $\boldsymbol{S}_B$. Given a set of $N$ feature vectors $\{\boldsymbol{x}_1, \boldsymbol{x}_2, \cdots, \boldsymbol{x}_N\} = \boldsymbol{X} \in \mathbb{R}^{N \times d}$, each feature $\boldsymbol{x}$ is associated with one of $C$ classes, the scatter matrices can be formulated as:

$$\boldsymbol{S}_W = \sum_{c=1}^{C} \sum_{\boldsymbol{x} \in \boldsymbol{X}_c} (\boldsymbol{x} - \bar{\boldsymbol{x}}_c)(\boldsymbol{x} - \bar{\boldsymbol{x}}_c)^T, \tag{3}$$

$$\boldsymbol{S}_B = \sum_{c=1}^{C} N_c (\bar{\boldsymbol{x}}_c - \bar{\boldsymbol{x}})(\bar{\boldsymbol{x}}_c - \bar{\boldsymbol{x}})^T, \tag{4}$$

$$\bar{\boldsymbol{x}} = \frac{1}{N} \sum_{i=1}^{N} \boldsymbol{x}_i, \qquad \bar{\boldsymbol{x}}_c = \frac{1}{N_c} \sum_{\boldsymbol{x} \in \boldsymbol{X}_c} \boldsymbol{x}, \tag{5}$$

where $\boldsymbol{X}_c$ represents the subset of $\boldsymbol{X}$ pertaining to class $c$. The optimal projection matrix, $\boldsymbol{W} \in \mathbb{R}^{d \times d'}$, is found by maximizing the Fisher criterion (Fisher, 1938), which is the ratio of the between-class scatter to the within-class scatter in the projected space:

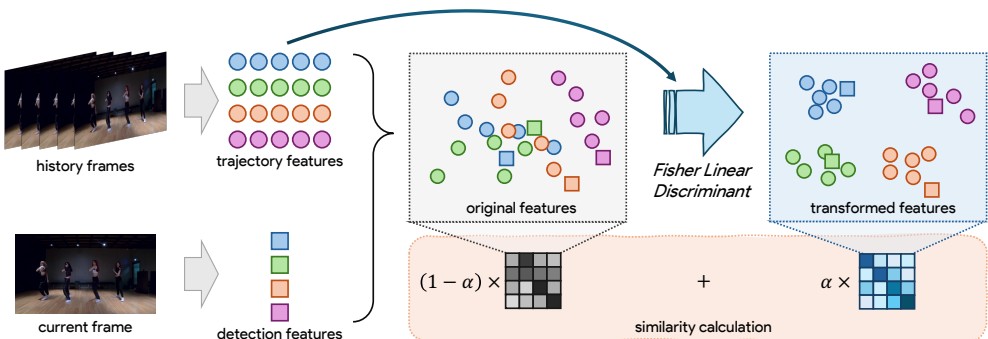

Figure 3: **Overview of our pipeline.** We use different colors to indicate different identities (trajectories). In the original space, some overly similar targets cannot be well distinguished, leading to issues in the matching process. Therefore, we treat the trajectory features as conditions and apply a tailored *Fisher Linear Discriminant* to seek a better subspace for distinguishing different trajectories. Finally, both original and transformed features are used to calculate the similarity matrix, balancing generalization and specialization.

$$J(\boldsymbol{W}) = \frac{\boldsymbol{W}^T \boldsymbol{S}_B \boldsymbol{W}}{\boldsymbol{W}^T \boldsymbol{S}_W \boldsymbol{W}}. \tag{6}$$

By applying the projection matrix $\boldsymbol{W}$ derived above, each feature $\boldsymbol{x}$ is converted into a new $d'$-dimensional vector with enhanced discriminability, where $d' = \min(C - 1, d)$.

## 3 METHOD

Based on the analysis in Section 2.2 and the result shown in Figure 2a, a clear positive correlation exists between the discriminative capability of the ReID features and the final tracking performance. Therefore, in this section, our primary goal is to find a more discriminative representation space for distinguishing between different trajectories. To this end, we mainly employ Fisher Linear Discriminant (FLD) (Fisher, 1938) along with several customized techniques, which are detailed in Section 3.1 and Sections 3.2 - 3.3, respectively. The overall illustration is shown in Figure 3.

### 3.1 HISTORY-AWARE TRANSFORMATION FOR REID FEATURES

As discussed in Section 1, current multi-object tracking (MOT) methods (Maggiolino et al., 2023; Yang et al., 2023b; Lv et al., 2024) largely adopt ReID features directly from traditional re-identification methods (Ristani & Tomasi, 2018; Luo et al., 2019). Since these models are required to distinguish between a vast number of open-set identities, the features they produce are, by design, as general as possible. In contrast, the multiple object tracking task only requires recognizing a closed set of identities within a single video. This creates a dilemma where the generality of traditional ReID features becomes a liability, as they lack the specificity needed to differentiate between these similar targets, as illustrated in Figure 1. Therefore, we are motivated to seek a more specialized representation space to address the aforementioned challenges. Intuitively, this space should pull features belonging to the same trajectory closer, while pushing features from different trajectories further apart. This idea coincides perfectly with the objective of Fisher Linear Discriminant (FLD) (Fisher, 1938) in its mathematical formulation, provided that we treat each *trajectory* as a *class* in the original framework. Specifically, by replacing the feature vector $\boldsymbol{x}$ in Equation 3 - 5 with our ReID features $\boldsymbol{f}$, and substituting the number of classes $C$ with the number of tracks $N_\tau$, we can obtain the projection matrix $\boldsymbol{W}$ for tracking by maximizing the objective in Equation 6.

However, FLD is a supervised method, which means it requires corresponding labels in addition to the feature vectors. This core prerequisite is unfulfilled in a standard tracking process. Therefore, we propose a history-aware dynamic labeling scheme to compensate for this absence. Practically,

since tracking is an online process, at each timestep $t$, the historical track assignments from previous frames can serve as the supervisory signals for FLD. Although potential tracking errors exist, we believe the overall statistical signal remains reliable. Furthermore, since a target's appearance gradually evolves during tracking, we only consider its $T$ most recent features for each trajectory. This choice ensures both efficiency and effectiveness.

## 3.2 TEMPORALLY-WEIGHTED TRAJECTORY CENTROID

Following the statement in Section 3.1, a naive implementation would be to average all $T$ features $\{\boldsymbol{f}_{t-T,\tau_j}, \cdots, \boldsymbol{f}_{t-2,\tau_j}, \boldsymbol{f}_{t-1,\tau_j}\}$ of the $\tau_j$-th trajectory to serve as its mean feature center. According to the definition of FLD (Fisher, 1938) and Equation 4, these feature centroids determine the distribution centers of the vectors after projection. Although this approach yields notable improvements, we still point out that it overlooks the temporal characteristics inherent in the tracking task. In online tracking, a target's appearance evolves continuously over time. Even within the same trajectory, features that are closer temporally tend to have higher similarity. Hence, for identity allocation at the current moment, more recent ReID features should intuitively play a more significant role. In practice, we apply a temporal weighting to the mean calculation in Equation 5:

$$\bar{\boldsymbol{f}} = \frac{1}{N_\tau} \sum_{j=1}^{N_\tau} \bar{\boldsymbol{f}}_{\tau_j}, \qquad \bar{\boldsymbol{f}}_{\tau_j} = \frac{1}{\sum \lambda_{t'}} \sum_{t'=t-T}^{t-1} \lambda_{t'} \boldsymbol{f}_{t',\tau_j}, \qquad \lambda_{t'} = (\lambda_0)^{t-t'}, \qquad (7)$$

where $\lambda_0$ is a temporal decay coefficient with a value between 0 and 1. Using these temporal-weighted trajectory centroids in the calculation of Equation 4 makes the final projection more attuned to the current temporal context, benefiting the similarity measurement at the time step $t$.

## 3.3 KNOWLEDGE INTEGRATION

Although we have found a more discriminative space conditioned on historical trajectories with the methods in Section 3.1 and 3.2, it still has some imperfections. First, the historical tracking results may contain errors, which can lead to a biased or suboptimal projection matrix. Second, because the transformed space is built only from the features of existing trajectories, it may not be robust enough for handling newborn targets. Therefore, we revisit the original representation space. Although it is not optimized for a given scenario, it offers more robust generalization capabilities, especially when facing unseen targets. This motivates our proposal to integrate it with the specialized subspace for a trade-off. Due to the disparate dimensionalities of these two spaces, our integration strategy operates on the similarity matrices rather than the vectors themselves. It can be formulated as follows:

$$\text{Cost}^*(t, i, \tau_j) = 1 - \text{Sim}^*(t, i, \tau_j) = 1 - [\alpha \cdot \text{Sim}'(t, i, \tau_j) + (1 - \alpha) \cdot \text{Sim}(t, i, \tau_j)], \quad (8)$$

where $\text{Sim}'(\cdot)$ is the similarity computed using the transformed ReID features, and $\alpha$ is a balancing coefficient. The Hungarian algorithm then finds the optimal assignment using the complete cost matrix constructed from the fused $\text{Cost}^*(\cdot)$. See Figure 3 for an overview of this pipeline.

## 4 EXPERIMENTS

### 4.1 DATASETS AND METRICS

**Datasets.** We select DanceTrack (Sun et al., 2022) and SportsMOT (Cui et al., 2023) as our primary experimental benchmarks because they both present a key challenge: targets within a single video often exhibit a high degree of visual similarity. Specifically, DanceTrack features group dance scenarios, while SportsMOT includes three types of team sports. We also evaluate our approach on the TAO (Dave et al., 2020) dataset to demonstrate its effectiveness in diverse and general tracking cases. In addition, we present the results on MOT17 (Milan et al., 2016) in Section C.1.

**Metrics.** On traditional MOT benchmarks (Milan et al., 2016; Sun et al., 2022; Cui et al., 2023), we select the Higher Order Tracking Accuracy (HOTA) (Luiten et al., 2021) as the primary metric, especially its Association Accuracy (AssA) component. We also include MOTA (Bernardin & Stiefelhagen, 2008) and IDF1 (Ristani et al., 2016) in some experiments. To better evaluate the

multi-category tracking problem, we employ the Tracking Every Thing Accuracy (TETA) (Li et al., 2022) on the TAO dataset (Dave et al., 2020).

## 4.2 IMPLEMENTATION DETAILS

To more clearly illustrate the improvements brought by our method, we focus our experiments on pure ReID-based trackers, as discussed in Section 2.1. Due to the lack of such publicly available trackers in the community, we construct a new tracker by combining the widely-used YOLOX (Ge et al., 2021) detector with the FastReID (Luo et al., 2019) model. For a fair comparison, we use the well-trained weights from Cao et al. (2022); Yang et al. (2023b); Lv et al. (2024) for all network modules. To ensure the baseline achieves its best performance, we optimize its hyperparameters on every benchmark via grid search. The resulting tracker is denoted as *FastReID-MOT*. As for MASA (Li et al., 2024a), we also bring the model weights from the official repository. For notation, we add the prefix *HAT-* to methods that use our **H**istory-**A**ware **T**ransformation approach.

Table 1: Performance comparison with state-of-the-art methods on the Dancetrack test set.

| Methods | HOTA | DetA | AssA |
|---|---|---|---|
| *motion-based:* | | | |
| ByteTrack (Zhang et al., 2022a) | 47.7 | 71.0 | 32.1 |
| DiffusionTrack (Luo et al., 2024) | 52.4 | 82.2 | 33.5 |
| OC-SORT (Cao et al., 2022) | 55.1 | 80.3 | 38.3 |
| C-BIoU (Yang et al., 2023a) | 60.6 | 81.3 | 45.4 |
| *ReID-based:* | | | |
| QDTrack (Pang et al., 2021) | 54.2 | 80.1 | 36.8 |
| FastReID-MOT (our baseline) | 50.6 | 81.1 | 31.6 |
| HAT-FastReID-MOT | 58.6 | 81.3 | 42.3 |
| HAT-FastReID-MOT† | **61.2** | **81.6** | **46.0** |
| *hybrid-based:* | | | |
| FairMOT (Zhang et al., 2021) | 39.7 | 66.7 | 23.8 |
| DeepSORT (Wojke et al., 2017) | 45.6 | 71.0 | 29.7 |
| StrongSORT (Du et al., 2023) | 55.6 | 80.7 | 38.6 |
| DiffMOT (Lv et al., 2024) | 62.3 | **82.5** | 47.2 |
| Hybrid-SORT-ReID (Yang et al., 2023b) | 65.7 | – | – |
| ByteTrack-ReID | 52.4 | 71.0 | 38.7 |
| HAT-ByteTrack-ReID | 56.1 | 71.4 | 44.2 |
| OC-SORT-ReID | 60.8 | 81.0 | 45.7 |
| HAT-OC-SORT-ReID | 64.6 | 81.5 | 51.3 |
| HAT-Hybrid-SORT-ReID | **66.9** | 81.5 | **55.0** |

Table 2: Performance on the SportsMOT test set. Gray results denote joint training involving the validation set of SportsMOT.

| Methods | HOTA | DetA | AssA |
|---|---|---|---|
| *motion-based:* | | | |
| ByteTrack (Zhang et al., 2022a) | 62.8 | 77.1 | 51.2 |
| OC-SORT (Cao et al., 2022) | 71.9 | 86.4 | 59.8 |
| ByteTrack (Zhang et al., 2022a) | 64.1 | 78.5 | 52.3 |
| OC-SORT (Cao et al., 2022) | 73.7 | 88.5 | 61.5 |
| *ReID-based:* | | | |
| QDTrack (Pang et al., 2021) | 60.4 | 77.5 | 47.2 |
| FastReID-MOT (our baseline) | 67.3 | 86.8 | 52.3 |
| HAT-FastReID-MOT | 78.1 | 87.3 | 69.9 |
| HAT-FastReID-MOT† | **78.9** | **87.4** | **71.3** |
| HAT-FastReID-MOT† | 80.8 | 89.4 | 73.1 |
| *hybrid-based:* | | | |
| BoT-SORT (Aharon et al., 2022) | 68.7 | 84.4 | 55.9 |
| DiffMOT (Lv et al., 2024) | 72.1 | 86.0 | 60.5 |
| ByteTrack-ReID | 65.1 | 76.8 | 55.1 |
| HAT-ByteTrack-ReID | 72.4 | 77.3 | 67.8 |
| OC-SORT-ReID | 74.1 | 86.8 | 63.3 |
| HAT-OC-SORT-ReID | **81.2** | **87.2** | **75.6** |
| HAT-OC-SORT-ReID | 82.4 | 89.3 | 76.1 |

## 4.3 STATE-OF-THE-ART COMPARISON

**FastReID-MOT.** We compare our method (HAT-FastReID-MOT) against the baseline (FastReID-MOT) on DanceTrack (Sun et al., 2022) and SportsMOT (Cui et al., 2023) in Table 1 and 2. † indicates that hyperparameters are fine-tuned on the corresponding dataset to maximize performance; otherwise, the default settings from our ablation study are used, as stated in Section 4.4. Our approach yields substantial performance gains over the baseline. On the challenging DanceTrack dataset, our appearance-only method even achieves results comparable to several recent hybrid and motion-based trackers (Yang et al., 2023a; Lv et al., 2024). Even more impressively, our ReID-only tracker establishes a new state-of-the-art, notably outperforming existing methods, including Lv et al. (2024), which shares the same ReID model. This result both vindicates our approach and highlights the need to reconsider the true potential of ReID features for target association.

**Hybrid-based Tracker.** To further validate the effectiveness of our method, we inserted it into several recent well-known trackers (Zhang et al., 2022a; Cao et al., 2022; Yang et al., 2023b). The results in Table 1 and 2 show that our method can consistently bring significant improvements when applied to hybrid-based trackers. The combination of our method with (Yang et al., 2023b) surpasses

Table 3: Evaluating our method with MASA (Li et al., 2024a). All models are trained on a large-scale image segmentation dataset (Kirillov et al., 2023) with different visual backbones.

| Methods | DanceTrack test | | SportsMOT test | | TAO val | |
|---|---|---|---|---|---|---|
| | HOTA | AssA | HOTA | AssA | TETA | AssocA |
| *MASA (Li et al., 2024a):* | | | | | | |
| MASA-R50 | 50.8 | 31.6 | 71.6 | 58.9 | 45.8 | 42.7 |
| MASA-Detic | 50.6 | 31.5 | 72.2 | 60.1 | 46.5 | 44.5 |
| MASA-G-DINO | 50.4 | 31.2 | 72.8 | 61.0 | 46.8 | 45.0 |
| MASA-SAM-B | 49.4 | 29.9 | 71.9 | 59.5 | 46.2 | 43.7 |
| *Ours:* | | | | | | |
| HAT-MASA-R50 | 54.3 (+3.5) | 36.1 (+4.5) | 73.7 (+2.1) | 62.4 (+3.5) | 46.4 (+0.6) | 44.4 (+1.7) |
| HAT-MASA-Detic | 54.3 (+3.7) | 36.2 (+5.7) | 74.5 (+2.3) | 63.7 (+3.6) | 47.2 (+0.7) | 46.4 (+1.9) |
| HAT-MASA-G-DINO | 53.9 (+3.5) | 35.7 (+4.5) | 74.7 (+1.9) | 64.1 (+3.1) | 47.5 (+0.7) | 46.7 (+1.7) |
| HAT-MASA-SAM-B | 52.1 (+2.7) | 33.4 (+3.5) | 73.4 (+1.5) | 61.9 (+2.4) | 46.9 (+0.7) | 45.6 (+1.9) |

Table 4: Comparison of different transformation selections. *Oracle* and *YOLOX* denote the sources of the detection results, while $d$ and $d'$ indicate the original and projected feature dimension, respectively. $N_{obj}$ and $N_{id}$ are the total number of historical samples and trajectories, respectively. If $d' > d$, the target dimension will be set to $d$.

| # | $\mathcal{D}$ | *Method* | $d'$ | DanceTrack val | | | SportsMOT val | | |
|---|---|---|---|---|---|---|---|---|---|
| | | | | HOTA | AssA | IDF1 | HOTA | AssA | IDF1 |
| # 1 | | – | $d$ | 74.9 | 57.3 | 72.0 | 86.2 | 74.7 | 84.0 |
| # 2 | *Oracle* | PCA | $N_{obj} - 1$ | 75.1 (+0.2) | 57.6 (+0.3) | 72.2 (+0.2) | 85.6 (-0.6) | 73.9 (-0.8) | 83.6 (-0.4) |
| # 3 | | PCA | $N_{id} - 1$ | 56.3 (-18.6) | 32.5 (-24.8) | 50.4 (-21.6) | 69.4 (-16.8) | 49.0 (-25.7) | 66.4 (-17.6) |
| # 4 | | FLD | $N_{id} - 1$ | **79.0 (+4.1)** | **63.8 (+6.5)** | **77.0 (+5.0)** | **92.2 (+6.0)** | **85.4 (+10.7)** | **90.7 (+6.7)** |
| # 5 | | – | $d$ | 51.1 | 33.4 | 51.0 | 73.7 | 61.5 | 76.6 |
| # 6 | *YOLOX* | PCA | $N_{obj} - 1$ | 45.3 (-5.8) | 26.7 (-6.7) | 40.0 (-11.0) | 70.6 (-3.1) | 56.4 (-5.1) | 71.5 (-5.1) |
| # 7 | | PCA | $N_{id} - 1$ | 43.5 (-7.6) | 24.6 (-8.8) | 39.4 (-11.6) | 61.3 (-12.4) | 42.7 (-18.8) | 61.4 (-15.2) |
| # 8 | | FLD | $N_{id} - 1$ | **57.7 (+6.6)** | **42.7 (+9.3)** | **56.7 (+5.7)** | **81.1 (+7.4)** | **74.1 (+12.6)** | **85.5 (+8.9)** |

all existing approaches and achieves the state-of-the-art performance (66.9 HOTA). The smaller performance gains on hybrid-based methods can be attributed to both performance saturation and the inherent design of these trackers, which often prioritizes motion and thus limits the impact of our appearance enhancements. Moreover, intricate algorithmic designs make inter-module harmonization challenging.

**MASA.** To investigate the generalization of our method across different ReID representation spaces, we conducted experiments on MASA (Li et al., 2024a). This framework is ideal as it includes a variety of visual backbones (He et al., 2016; Zhou et al., 2022; Liu et al., 2024; Kirillov et al., 2023) and is pre-trained on a general-purpose segmentation dataset (Kirillov et al., 2023). The TAO (Dave et al., 2020) benchmark is introduced to serve as a general-purpose tracking scenario. Table 3 shows that our approach brings consistent and significant boosts across all tested visual backbones. However, the improvements are more minor compared to Table 1 and 1. We argue that our method's fundamental property is to refine an existing feature space, but since MASA is not trained with tracking datasets, its representations lack the specific discriminative capability needed for our approach to distill. Furthermore, the TAO dataset (Dave et al., 2020) contains numerous object categories with low similarity to one another, which also limits the applicability of our algorithm.

## 4.4 ABLATION STUDY

We verify the effectiveness of each component in this section, using the ReID-based tracker from Section 2.1 and 4.2 as the baseline. For all experiments, except those in Table 4, we use the detections from the public YOLOX model (Ge et al., 2021; Cao et al., 2022). The experimental results are shown incrementally, with each table adding one component at a time. For the hyperparameter explorations, the gray background indicates the default settings we determined through experiments.

**History-Aware Transformation.** As shown in Table 4, applying the FLD-based ReID feature transformation, as described in Section 3.1, significantly improves tracking performance. Following the correlation analysis in Section 2.2, we visualize the change in the discriminative ability $\delta_{1000}$ of the ReID features under an oracle detection setting, as shown in Figure 2b. This serves as clear evidence that our history-aware transformation boosts the separability of visual representations, thereby improving tracking capabilities. For comparison, we also evaluate a PCA-based transformation in Table 4, but it resulted in a performance drop. This is because Principal Component Analysis (PCA) is designed to maximize global data variance and is oblivious to the trajectory labels, which we believe are essential for finding an optimal representation for tracking.

Table 5: Exploration of the history length $T$.

| $T$ | HOTA | AssA | MOTA | IDF1 |
|---|---|---|---|---|
| 10 | 54.3 | 37.8 | **86.7** | 53.2 |
| 20 | 56.2 | 40.4 | 86.5 | 55.2 |
| 40 | 57.0 | 41.4 | 86.5 | 56.6 |
| 60 | **57.7** | **42.7** | 86.3 | **56.7** |
| 80 | 56.3 | 40.5 | 86.2 | 55.4 |
| $\infty$ | 55.3 | 39.2 | 85.1 | 52.2 |

Table 6: Effect of the temporal decay coefficient $\lambda_0$.

| $\lambda_0$ | HOTA | AssA | MOTA | IDF1 |
|---|---|---|---|---|
| 1.00 | 57.7 | 42.7 | 86.3 | 56.7 |
| 0.95 | 58.5 | 42.5 | 86.8 | 58.2 |
| 0.90 | **59.3** | **44.8** | 86.9 | **59.8** |
| 0.80 | 59.1 | 44.7 | **87.0** | 59.6 |
| 0.60 | 58.1 | 43.2 | **87.0** | 57.9 |
| 0.40 | 57.8 | 42.8 | 86.9 | 57.1 |

**History Length $T$.** As stated at the end of Section 3.1, we only consider ReID features from the $T$ most recent frames. Although using a too-short temporal length $T$ decreases the credibility of the reference samples, it still provides a notable enhancement compared to the baseline tracker (54.3 *vs.* 51.1 HOTA), as shown in Table 5. Conversely, a too large $T$ would incorporate outdated features, making the distribution less representative of the current state and ultimately harming performance.

Table 7: Analysis of the balancing coefficient $\alpha$.

| $\alpha$ | HOTA | AssA | MOTA | IDF1 |
|---|---|---|---|---|
| 1.0 | 59.3 | 44.8 | 86.9 | 59.8 |
| 0.9 | **60.6** | **46.8** | 87.1 | **61.7** |
| 0.8 | 59.3 | 45.0 | 87.2 | 60.4 |
| 0.6 | 59.1 | 44.7 | **87.6** | 60.6 |

**Temporally-Weighted Trajectory Centroid.** Following our discussion in Section 3.2 about the varying temporal importance of features, we introduce the coefficient $\lambda_0$ to weight them accordingly. Experimental results in Table 6 demonstrate that using the temporal-weighted trajectory centroid can significantly enhance tracking performance. However, it is essential to note that excessively small values of $\lambda_0$ may lead to an overreliance on recent samples, resulting in a decline in robustness.

**Knowledge Integration.** In Table 7, we investigate various fusion coefficients $\alpha$ to balance robustness and specialization. These results indicate that this is a trade-off art, prompting us to choose 0.9 as our default setting. In addition, this supports the concept outlined in Section 3.3, valuing the complementarity of those two spaces can boost the reliability of ReID features.

## 5 CONCLUSION

In this paper, we challenge a long-standing practice in multiple object tracking (MOT): *the direct adoption of appearance matching strategies from the re-identification task, an approach we argue is fundamentally inappropriate for tracking*. We contend that visual representations in MOT should be tailored to discriminate among the finite set in a given video sequence, as opposed to the open-set challenge. To this end, we proposed an approach that leverages the tracking history to guide an adaptive transformation of the feature space, thereby boosting its discriminability. Comprehensive experiments validate the effectiveness and versatility of our proposed approach and establish the new state-of-the-art performance. These results serve as compelling evidence that the potential of ReID features in MOT has been significantly underestimated. Therefore, we hope our findings spur a wave of research into this crucial problem, whether in the form of new training-free components or as guiding principles for developing learnable modules.

REPRODUCIBILITY STATEMENT

As stated in Section 4.2, all model weights used in our experiments are directly borrowed from public repositories (Cao et al., 2022; Yang et al., 2023b; Lv et al., 2024). Our dataset organization and evaluation procedures are all conducted using peer-reviewed and publicly available methodologies and code (Milan et al., 2016; Jonathon Luiten, 2020; Sun et al., 2022; Cui et al., 2023; Li et al., 2024a; Gao et al., 2025). To guarantee reproducibility, we will open-source the code for our final experiments and the corresponding tracker results.

THE USE OF LARGE LANGUAGE MODELS

We used Large Language Models (LLMs) for assistance with translating, polishing, and correcting the grammar of the text in this paper, as well as for generating formatted LATEX code. We have also utilized the LLM assistance in some of the visualization code.

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

# A  RELATED WORK

**Tracking-by-Detection** methods decouple the multiple object tracking (MOT) task into two sub-tasks: object detection and data association. While a minority of studies (Khurana et al., 2021) have explored customized detection methods, the vast majority of research (Zhou et al., 2020; Zhang et al., 2022a; Mancusi et al., 2023; Liu et al., 2023; Saraceni et al., 2024) has focused on the design of the target association algorithm. In this process, researchers model trajectories and measure affinities based on diverse cues. For visual appearance, most methods (Aharon et al., 2022; Maggiolino et al., 2023; Yang et al., 2023b; Lv et al., 2024) directly utilize off-the-shelf ReID models (Luo et al., 2019) to extract features. While some approaches (Zhang et al., 2021; Wang et al., 2020; Plaen et al., 2024) employ custom-designed extractors, they still adhere to the fundamental principles and supervision methods of traditional ReID methods (Ristani & Tomasi, 2018; Luo et al., 2019). Regarding location information, the most classic method (Bewley et al., 2016; Zhang et al., 2022a) is to use the Kalman filter (Welch et al., 1995) for linear estimation of the motion. To handle non-linear dynamics (Sun et al., 2022; Cui et al., 2023) and other complex cases, recent methods have introduced many tailored rules (Cao et al., 2022; Du et al., 2023; Yang et al., 2023b; Yi et al., 2024) or adopted learnable modules for motion prediction (Dendorfer et al., 2022; Qin et al., 2023; Luo et al., 2024; Lv et al., 2024; Xiao et al., 2024; Huang et al., 2024). Many approaches (Wojke et al., 2017; Du et al., 2023; Maggiolino et al., 2023; Yang et al., 2023b; Lv et al., 2024) also fuse the two aforementioned cues together to fully leverage their respective advantages. Furthermore, some other methods introduce even more information modalities, such as Bird's-Eye-View (BEV) perspectives (Dendorfer et al., 2022) and depth information (Aharon et al., 2022; Mancusi et al., 2023; Wang et al., 2025). Most relevant to our work are several studies that aim to customize the ReID branch of the MOT task. Hou et al. (2022) seeks to mitigate the mismatch between its global temporal training and local temporal inference, Chen et al. (2024) performs group-wise similarity calculation to address the long-tail distribution problem, Li et al. (2024b) helps newborn targets acquire more robust representations, Cao et al. (2025) sharpens the distinction in similarity. These methods do not focus on the discriminability of the representation space or leverage the information difference between intra- and inter-trajectory data. Therefore, they differ from our method in both core philosophy and primary contribution.

**End-to-End MOT** models are emerging forces, bypassing hand-crafted algorithms (Zhang et al., 2022a; Cao et al., 2022) to formulate multi-object tracking in an end-to-end manner (Zeng et al., 2022; Gao et al., 2025). A typical form is to expand DETR (Carion et al., 2020; Zhu et al., 2021) into MOT tasks, representing different trajectories through the propagation of track queries (Zeng et al., 2022; Meinhardt et al., 2022). Follow-up methods incorporated temporal information (Cai et al., 2022; Gao & Wang, 2023; Segù et al., 2024) and mitigated the imbalance of supervision signals (Zhang et al., 2022b; Yan et al., 2025), leading to better tracking performance. Nevertheless, end-to-end methods still face the challenges of high computational costs and a strong need for training data, which will require future research.

# B  EXPERIMENTAL DETAILS

## B.1  ReID FEATURE DISCRIMINATIVE CAPABILITY

As stated in Section 2.2, we adopt the metric $\delta_{1000}$ to quantify the discriminative capability of the representation space. This metric is derived from individual discriminative scores that are computed for each detection. Formally, for the $i$-th detection at time step $t$, we calculate the similarities against all history trajectories, as specified in Equation 2. A discriminative score $\delta(t, i)$ for this detection is then defined as:

$$\delta(t, i) = \text{Sim}^+(t, i, \tau_j) - \max_j \big[\text{Sim}^-(t, i, \tau_j)\big], \tag{9}$$

where $\text{Sim}^+(t, i, \tau_j)$ denotes the similarity to the corresponding positive sample (the $i$-th detection belongs to the $j$-th trajectory), and $\text{Sim}^-(t, i, \tau_j)$ denotes the similarity to a negative sample. We select the most similar negative sample using $\max_j \big[\text{Sim}^-(t, i, \tau_j)\big]$, because the most confusing example directly determines whether a misallocation of identities will occur.

After calculating all valid $\delta(t, i)$ within a video sequence, we aggregate them to obtain the overall discriminative measure. Since tracking errors like ID switches occur in a very small portion of a long video (thousands of frames), we select the 1000 most challenging cases from all discriminative scores. In practice, we sort the scores in ascending order and select the smallest 1000 samples to compute the averaging score $\delta_{1000}$, since these items are the most likely to be misassigned in tracking.

## B.2 ReID-based Tracker: FastReID-MOT

As stated in Section 2.1 and 4.2, our baseline FastReID-MOT relies solely on ReID features for tracking. To keep the baseline straightforward, we implement a single-stage online tracker with a minimal set of hyperparameters:

- $\lambda$, the feature update ratio in Equation 1.
- $\theta_{\text{det}}$, detections with a confidence exceeding this threshold are considered by the tracker.
- $\theta_{\text{sim}}$, identity assignments with a similarity score exceeding this threshold are considered as valid choices.
- $\theta_{\text{new}}$, unmatched detections with a confidence exceeding this threshold are considered as newborn targets.
- $\theta_{\text{miss}}$, a trajectory is terminated if the number of consecutive missing frames is greater than this threshold.

All the aforementioned hyperparameters are tuned using a grid search on the corresponding datasets to maximize the baseline's performance. In subsequent ablation experiments, we do not adjust these hyperparameters to ensure that the observed improvements are purely attributable to our proposed method.

## B.3 MASA Details

In the MASA (Li et al., 2024a) inference process, we simplified the original bi-softmax matching procedure Li et al. (2024a); Pang et al. (2021) to the simple cosine similarity combined with the Hungarian algorithm (as we detailed in Section 2.1 and Equation 2), and tuned some hyperparameters, which resulted in a slight improvement in tracking performance across all datasets. For our hyperparameters, we primarily adhered to the default settings outlined in Section 4.4, with the exception of adjusting $\alpha$ to 0.5 to better accommodate MASA's feature representation.

## B.4 Oracle Setting

In Table 4 and Figure 2, we leverage an *oracle setting* to focus our analysis on tracking performance without the influence of other factors. In these experiments, we use the bounding boxes' coordinates from the ground truth files as the detection results and set all confidence scores to 1.0. Even under these ideal conditions, the Detection Accuracy (DetA) will not reach 100.0, as a result of the metric's calculation method (Luiten et al., 2021).

## B.5 Ablation Study

As we stated in Section 4.4, the ablation experiments are conducted incrementally, with each table adding one component at a time:

- In Table 4, we apply $T = 60$, $\lambda_0 = 1.0$ and $\alpha = 1.0$, which means we do not use the *temporally-weighted trajectory centroid* and *knowledge integration*.
- In Table 5, we apply $\lambda_0 = 1.0$ and $\alpha = 1.0$, which means we do not use the *temporally-weighted trajectory centroid* and *knowledge integration*.
- In Table 6, we apply $T = 60$ and $\alpha = 1.0$, which means we do not use the *knowledge integration*.

Table 8: Performance comparison with state-of-the-art methods on MOT17 (Milan et al., 2016). The best and second-best results are denoted in **bold** and underline, respectively.

| Methods | HOTA | DetA | AssA | IDF1 |
|---|---|---|---|---|
| *motion-based:* | | | | |
| ByteTrack (Zhang et al., 2022a) | 63.1 | 64.5 | 62.0 | 77.3 |
| OC-SORT (Cao et al., 2022) | 63.2 | 63.2 | 63.4 | 77.5 |
| C-BIoU (Yang et al., 2023a) | 64.1 | 64.8 | 63.7 | 79.7 |
| *reid-based:* | | | | |
| QDTrack (Pang et al., 2021) | 53.9 | 55.6 | 52.7 | 66.3 |
| ContrasTR (Plaen et al., 2024) | 58.9 | – | – | 71.8 |
| FastReID-MOT (baseline) | 61.5 | 63.4 | 60.0 | 73.5 |
| HAT-FastReID-MOT† | **63.5** | **64.0** | **63.2** | **77.5** |
| *hybrid-based:* | | | | |
| FairMOT (Zhang et al., 2021) | 59.3 | 60.9 | 58.0 | 72.3 |
| DeepSORT (Wojke et al., 2017) | 61.2 | 63.1 | 59.7 | 74.5 |
| MixSort-OC (Cui et al., 2023) | 63.4 | 63.8 | 63.2 | 77.8 |
| DiffMOT (Lv et al., 2024) | **64.5** | **64.7** | **64.6** | **79.3** |
| OC-SORT-ReID | 64.1 | 64.4 | 64.0 | 79.0 |
| HAT-OC-SORT-ReID | 64.2 | 64.4 | 64.1 | 79.2 |

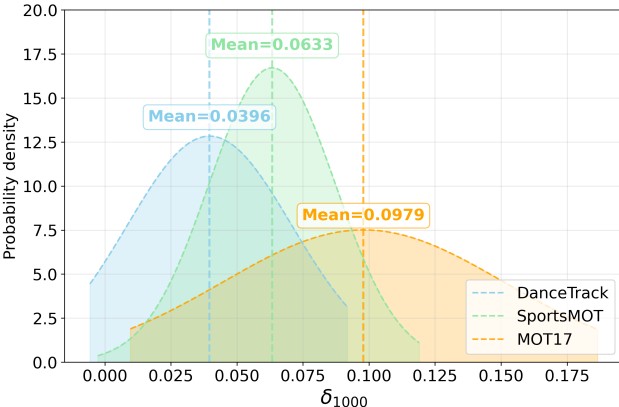

Figure 4: Comparison of ReID separability on DanceTrack (Sun et al., 2022), SportsMOT (Cui et al., 2023), and MOT17 (Milan et al., 2016) based on $\delta_{1000}$.

- In table 7, we apply $T = 60$ and $\lambda_0 = 0.9$, which means both proposed components are used in these experiments.

Together, these settings make up our default configuration ($T = 60$, $\lambda_0 = 0.9$, $\alpha = 0.9$) and are applied uniformly to all datasets as the default, as mentioned in Section 4.3 and 4.4.

### B.6 VISUALIZATION OF REID FEATURES

To qualitatively evaluate the discriminative capability of ReID features, we visualize feature similarities both within and across sequences. In Figure 1a, we show the features of objects in 15 consecutive frames of a single video sequence, projected to a two-dimensional space using Principal Component Analysis (PCA). In Figure 1b, we randomly select 10 sequences from the DanceTrack dataset (Sun et al., 2022) and visualize features extracted from 40 consecutive frames of each sequence, also projected via PCA.

Table 9: Performance comparison with state-of-the-art methods on the MOT20 validation set. The best results are denoted in **bold**.

| Methods | HOTA | DetA | AssA | MOTA | IDF1 |
|---|---|---|---|---|---|
| Deep OC-SORT (Maggiolino et al., 2023) | 59.5 | – | 58.2 | – | 76.3 |
| Hybrid-SORT-ReID (Yang et al., 2023b) | 60.7 | 61.6 | 60.0 | 74.0 | 78.4 |
| FastReID-MOT | 57.7 | 61.7 | 54.1 | 74.5 | 72.4 |
| HAT-FastReID-MOT | **61.2** | **62.3** | **60.4** | **75.0** | **78.8** |

## C  MORE RESULTS

### C.1  MOT17

In Table 8, we present our experimental results on the MOT17 (Milan et al., 2016) dataset. Due to the submission limits of the MOT17 evaluation server, we built our hybrid-based tracking using only the classic OC-SORT (Cao et al., 2022) algorithm. Compared to our baseline (FastReID-MOT), our method yields a significant performance gain (2.0 HOTA and 3.2 AssA), though the margin is not as large as on other benchmarks (Sun et al., 2022; Cui et al., 2023). We attribute this to the fact that the MOT17 dataset, consisting solely of pedestrians, has high inherent target discriminability (*e.g.*, distinct clothing colors and styles), which limits the room for our method to make a greater impact. In the hybrid-based experiments, we do not achieve a highly satisfactory performance. On the one hand, prior studies (Zhang et al., 2022a; Yang et al., 2023a) have shown that the simple motion patterns within MOT17 allow the motion prediction module to take a dominant role, thereby constraining the influence of the ReID branch. Our observations in Figure 4, based on $\delta_{1000}$, also confirm this. The ReID features of MOT17 targets show significantly greater separability, despite the dataset containing up to ten times more targets per frame compared to DanceTrack (Sun et al., 2022) and SportsMOT (Cui et al., 2023). On the other hand, the overly engineered fusion of multiple modules and the unreliable validation set split further increased the difficulty of optimizing the entire method. Despite these challenges, we still outperform MixSort-OC (Cui et al., 2023) that also uses OC-SORT as the framework, and are slightly behind Lv et al. (2024), which is based on learnable motion estimation.

To summarize, although our method does not achieve flawless results on MOT17, the consistent performance gains across experiments robustly demonstrate its effectiveness and applicability in diverse scenarios. Coupled with its outstanding performance across various other scenarios (Sun et al., 2022; Cui et al., 2023; Dave et al., 2020) in Table 1, 2 and 3, our method still holds enough promise and is attractive for future exploration.

### C.2  MOT20

We also evaluate our method on MOT20 (Dendorfer et al., 2020). To ensure fairness, all algorithms are implemented using the same public FastReID (Luo et al., 2019) weight. As reported in Table 9, our proposed method consistently outperforms both advanced trackers and our FastReID-MOT baseline. These results demonstrate the effectiveness of the proposed history-aware feature transformation under crowded scenes, and further validate the generalization ability of our method.

### C.3  EXTENDED EVALUATION WITH ADDITIONAL METRICS

To provide a more comprehensive and fine-grained evaluation of tracking performance, we report an extended set of metrics in Table 10. These complementary metrics allow a more thorough assessment of detection accuracy, association robustness, and identity consistency.

### C.4  COMPARISON WITH END-TO-END METHODS

End-to-end (E2E) trackers and heuristic tracking-by-detection methods follow fundamentally different paradigms, which makes direct comparisons inherently unfair and scenario-dependent. To

Table 10: Detailed performance comparison with state-of-the-art methods on the Dancetrack test set. By default, higher values indicate better performance, while metrics marked with ↓ denote that lower values are better.

| Methods | HOTA | DetA | AssA | LocA | MOTA | IDF1 | IDR | IDP | IDTP | IDFN↓ | IDFP↓ |
|---|---|---|---|---|---|---|---|---|---|---|---|
| *motion-based:* | | | | | | | | | | | |
| ByteTrack (Zhang et al., 2022a) | 47.7 | 71.0 | 32.1 | - | 89.6 | 53.9 | - | - | - | - | - |
| DiffusionTrack (Luo et al., 2024) | 52.4 | 82.2 | 33.5 | - | 89.3 | 47.5 | - | - | - | - | - |
| OC-SORT (Cao et al., 2022) | 55.1 | 80.3 | 38.3 | - | 92.0 | 54.6 | - | - | - | - | - |
| C-BIoU (Yang et al., 2023a) | 60.6 | 81.3 | 45.4 | - | 91.6 | 61.6 | - | - | - | - | - |
| *ReID-based:* | | | | | | | | | | | |
| QDTrack (Pang et al., 2021) | 54.2 | 80.1 | 36.8 | - | 87.7 | 50.4 | - | - | - | - | - |
| FastReID-MOT (our baseline) | 50.6 | 81.1 | 31.6 | 92.5 | 90.3 | 50.4 | 48.6 | 52.4 | 140635 | 148531 | 127941 |
| HAT-FastReID-MOT | 58.6 | 81.3 | 42.3 | 92.6 | 89.6 | 57.9 | 55.7 | 60.4 | 161074 | 128092 | 105783 |
| HAT-FastReID-MOT† | **61.2** | **81.6** | **46.0** | **92.7** | **89.7** | **61.1** | 58.7 | 63.7 | 169663 | 119503 | 96884 |
| *hybrid-based:* | | | | | | | | | | | |
| FairMOT (Zhang et al., 2021) | 39.7 | 66.7 | 23.8 | - | 82.2 | 40.8 | - | - | - | - | - |
| DeepSORT (Wojke et al., 2017) | 45.6 | 71.0 | 29.7 | - | 87.8 | 47.9 | - | - | - | - | - |
| StrongSORT (Du et al., 2023) | 55.6 | 80.7 | 38.6 | - | 91.1 | 55.2 | - | - | - | - | - |
| DiffMOT (Lv et al., 2024) | 62.3 | **82.5** | 47.2 | - | **92.8** | 63.0 | - | - | - | - | - |
| Hybrid-SORT-ReID (Yang et al., 2023b) | 65.7 | - | - | - | 91.8 | 67.4 | - | - | - | - | - |
| ByteTrack-ReID | 52.4 | 71.0 | 38.7 | 85.1 | 87.9 | 60.4 | 58.2 | 62.7 | 168175 | 120991 | 99897 |
| HAT-ByteTrack-ReID | 56.1 | 71.4 | 44.2 | 85.1 | 88.5 | 65.7 | 63.6 | 68.0 | 183838 | 105328 | 86371 |
| OC-SORT-ReID | 60.8 | 81.0 | 45.7 | 92.4 | 90.6 | 63.5 | 61.2 | 65.9 | 177073 | 112093 | 91589 |
| HAT-OC-SORT-ReID | 64.6 | 81.5 | 51.3 | **92.6** | 90.3 | 67.7 | 65.1 | 70.4 | 188348 | 100818 | 79266 |
| HAT-Hybrid-SORT-ReID | **66.9** | 81.5 | **55.0** | 92.6 | 90.5 | **71.3** | 68.7 | 74.2 | 198722 | 90444 | **69211** |

Table 11: Performance comparison with end-to-end methods on the Dancetrack test set.

| Methods | HOTA | DetA | AssA |
|---|---|---|---|
| *end-to-end:* | | | |
| MOTR (Zeng et al., 2022) | 54.2 | 73.5 | 40.2 |
| MeMOTR (Gao & Wang, 2023) | 63.4 | 77.0 | 52.3 |
| CO-MOT (Yan et al., 2025) | 65.3 | 80.1 | 53.5 |
| SambaMOTR (Segù et al., 2024) | 67.2 | 78.8 | 57.5 |
| MOTIP (Gao et al., 2025) | **69.6** | **80.4** | **60.4** |
| *heuristic:* | | | |
| HAT-ByteTrack-ReID | 56.1 | 71.4 | 44.2 |
| HAT-FastReID-MOT | 58.6 | 81.3 | 42.3 |
| HAT-FastReID-MOT† | 61.2 | **81.6** | 46.0 |
| HAT-OC-SORT-ReID | 64.6 | 81.5 | 51.3 |
| HAT-Hybrid-SORT-ReID | **66.9** | 81.5 | **55.0** |

Table 12: Performance on the SportsMOT test set. Gray results denote joint training involving the validation set of SportsMOT.

| Methods | HOTA | DetA | AssA |
|---|---|---|---|
| *end-to-end:* | | | |
| TrackFormer (Meinhardt et al., 2022) | 63.3 | 66.0 | 61.1 |
| MeMOTR (Gao & Wang, 2023) | 68.8 | 82.0 | 57.8 |
| MOTIP (Gao et al., 2025) | **72.6** | **83.5** | **63.2** |
| *heuristic:* | | | |
| HAT-ByteTrack-ReID | 72.4 | 77.3 | 67.8 |
| HAT-FastReID-MOT | 78.1 | 87.3 | 69.9 |
| HAT-FastReID-MOT† | 78.9 | **87.4** | 71.3 |
| HAT-OC-SORT-ReID | **81.2** | 87.2 | **75.6** |
| HAT-FastReID-MOT† | 80.8 | 89.4 | 73.1 |
| HAT-OC-SORT-ReID | 82.4 | 89.3 | 76.1 |

provide a complete perspective, we nevertheless report comparisons with representative E2E methods on both DanceTrack and SportsMOT.

On DanceTrack, our method achieves a HOTA score of 66.9, which is competitive with recent E2E approaches, exceeding CO-MOT(Yan et al., 2025) and being comparable to SambaMOTR(Segù et al., 2024). On SportsMOT, our method significantly outperforms all existing published E2E trackers, e.g., 81.2 versus 72.6 of MOTIP(Gao et al., 2025), demonstrating a clear advantage. These results indicate that neither E2E nor heuristic method uniformly dominate across all datasets. Instead, their relative effectiveness is highly scenario-dependent. Our method demonstrates strong competitiveness against state-of-the-art E2E models on DanceTrack and achieves decisive superiority on SportsMOT, further validating the practical value and versatility of the proposed framework.

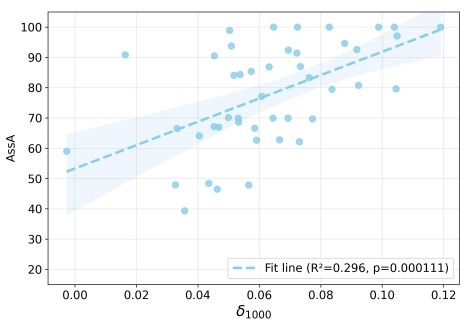 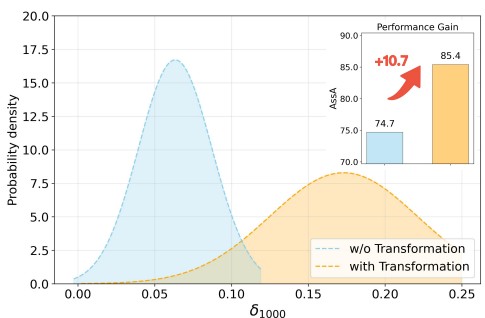

(a) Significant and reliable positive correlation between discriminability and tracking performance.

(b) Our transformation improves performance by enhancing feature discriminative capability.

Figure 5: Correlation between ReID feature discriminability $\delta_{1000}$ and tracking accuracy AssA on SportsMOT (Cui et al., 2023).

### C.5 INFERENCE SPEED

Given the detection results (without the latency of detectors), our method (including the ReID model (Luo et al., 2019)) achieves an inference speed of 22.7 FPS, compared to 46.5 FPS for the baseline, on DanceTrack (Sun et al., 2022) using an NVIDIA RTX A5000 GPU and an AMD Ryzen 9 5900X CPU. Although this meets the requirements for near real-time tracking, we must point out two main challenges that remain for achieving faster inference.

Based on our experiments, nearly all of the additional latency originates from the computation of eigenvalues and eigenvectors, as this operation is on the CPU (with `scipy.linalg.eigh(S_B, S_W)`), which is inherently inefficient for matrix calculations. We explored some alternative GPU-based packages like PyTorch, JAX, and CuPy. These packages offer CUDA acceleration for eigenvector computations (`eigh()` function). However, they lack an interface for generalized eigenvalue solving in `eigh()` (*e.g.*, discussed in #5461 issue[1] in the official repository of JAX, it only accepts one matrix for the eigenvalue calculation), which is a feature provided by SciPy and used for FLD solution. Transforming the input into a format acceptable for these functions incurs additional computational overhead and results in a loss of precision. If the same interface can be used, we estimate, based on experience, that it would result in a $4\times$ to $10\times$ speedup.

Moreover, the redundancy in feature dimensions further exacerbates this issue (2048 from FastReID (Luo et al., 2019) *vs.* 256 from MASA (Li et al., 2024a)), since latency increases with dimension count. This issue could be mitigated by either employing other dimensionality reduction methods or by reducing the output dimension of the ReID feature head during the training phase.

In summary, we consider that addressing this operator issue falls beyond the scope of this paper as it pertains to a complicated engineering problem.

### C.6 VISUALIZATIONS

#### C.6.1 DISCRIMINATIVE CAPABILITY ANALYSIS ON SPORTSMOT

As stated in Section 2.2, we observe a significant and reliable positive correlation between the discriminative capability ($\delta_{1000}$) of ReID features and the object association accuracy (AssA (Luiten et al., 2021)) on DanceTrack (Sun et al., 2022). To further examine the generality of this relationship, we extend the analysis to the SportsMOT dataset (Cui et al., 2023). As shown in Figure 5a, the visualizations on SportsMOT also demonstrate a consistently positive and statistically meaningful correlation between $\delta_{1000}$ and AssA, in agreement with the findings on DanceTrack in Figure 2a. It strongly supports our direction: improving discriminative capability to boost tracking performance.

---

[1] `https://github.com/jax-ml/jax/issues/5461`

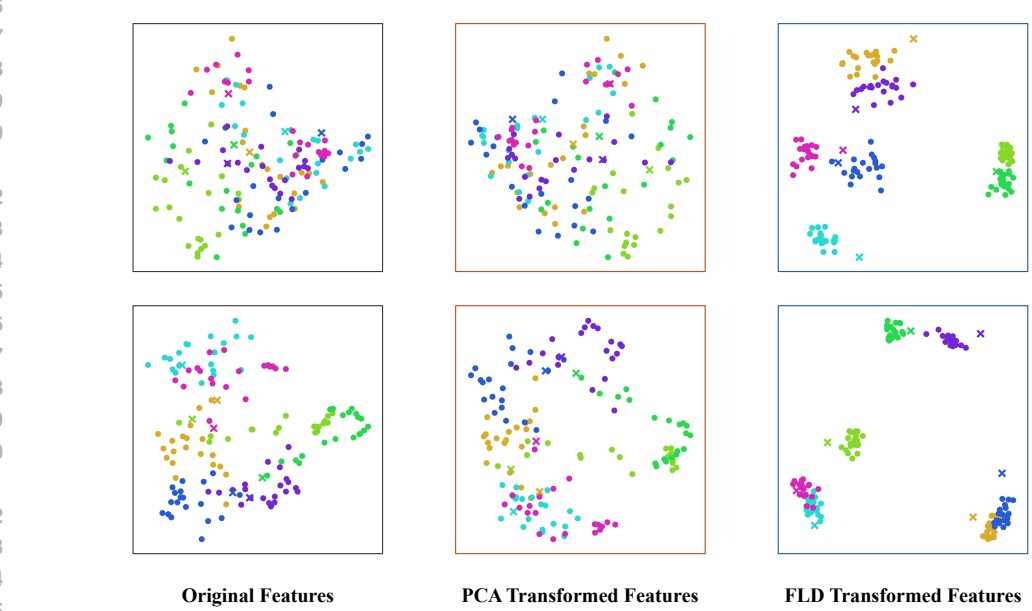

**Original Features**    **PCA Transformed Features**    **FLD Transformed Features**

Figure 6: **Visualization of ReID features.** ● represents the historical features and ✖ indicates the current features. Compared to the other two spaces, the FLD-projected space shows better differentiation of trajectories.

Figure 5b validates that our transformation can enhance feature discriminability to improve tracking performance on SportsMOT (Cui et al., 2023). This result, echoing the findings in Figure 2b, further substantiates our core hypothesis.

### C.7 VISUALIZATION OF ReID FEATURES

To further assess the impact of feature transformation on ReID discriminability, we visualize the features in different linear projection spaces in Figure 6. Features transformed by FLD exhibit clearer inter-trajectory separation than those produced by PCA or the original space. Taken together, the quantitative gains reported in Table 4 and the qualitative improvements observed in the visualizations indicate that incorporating historical trajectory information into the projection step is a principled and effective strategy for improving multiple object tracking: historical trajectories constitute an invaluable supervisory signal for representation selection and should therefore be exploited in the reasoning pipeline rather than disregarded.

## D LIMITATIONS

While our method has yielded encouraging results, there are some limitations and concerns that need to be pointed out.

**Hybrid-based Tracker.** While our method demonstrates significant improvements for ReID-based trackers, its gains on hybrid-based methods are somewhat limited. Besides the saturated metrics and overly complex algorithmic design discussed in Section 4.3, a deeper, more fundamental bias lies at the core: current hybrid-based trackers prioritize location information. For instance, in existing hybrid-based methods (Maggiolino et al., 2023; Yang et al., 2023b; Lv et al., 2024), the assignment stage often relies entirely or heavily on the IoU metric. This leads to the ReID information being either overlooked or not sufficiently trusted, thereby creating a disconnect between the ReID branch and performance improvement. Our method enhances the trustworthiness of ReID features, which may inspire future hybrid-based methods to develop ReID-first or more ReID-reliant trackers. We believe this could significantly alter the algorithmic logic of existing trackers, which we leave for future work to explore.

**End-to-End Method.** A potential concern is that our method cannot be applied to state-of-the-art end-to-end models (Segù et al., 2024; Yan et al., 2025; Gao et al., 2025). First, we argue that heuristic and end-to-end methods represent two distinct paths to the same goal, with no inherent superiority of one over the other, a common phenomenon in computer vision (Carion et al., 2020; Ge et al., 2021; Dhariwal & Nichol, 2021; Sun et al., 2024). Therefore, our proposed method does not need to compete directly with end-to-end approaches, and its inability to serve them is acceptable. This does not diminish the value of our method. Second, while our proposed history-aware transformation cannot be directly applied to end-to-end methods (*e.g.*, track queries), we believe it offers a valuable philosophical insight. Specifically, the observation that the information disparity between intra- and inter-trajectory features in historical tracklets can help a model better distinguish different tracks and thus improve tracking performance. This insightful conclusion might help guide the design of trainable or end-to-end models, which could potentially enable our ideas to extend beyond the realm of heuristic algorithms.

