# OpenReview forum: "History-Aware Transformation of ReID Features for Multiple Object Tracking"
_ICLR.cc/2026/Conference — Submitted to ICLR 2026_

### Official Review · Reviewer_zen2 · 2025-10-25

**Soundness:** 3
**Presentation:** 3
**Contribution:** 2
**Rating:** 4
**Confidence:** 4

**Summary:**

To achieve more robust ReID-based matching in MOT, the authors propose HAT to make ReID features more discriminative. Specifically, FLD transformation is applied to shrink intra-trajectory feature distance and enlarge inter-trajectory feature distance. Meanwhile, feature centroid update strategy is designed to prioritize features from recent observations.

**Strengths:**

1) Application of FLD transformation for ReID feature can be directly applied to other TBD trackers, and inspires more in-depth studies on other transformation techniques to improve ReID feature discriminativeness.
2) The proposed feature centroid update strategy provides a alternative for the widely-used EMA strategy.
3) the manuscript is well-organized with clear expressions.

**Weaknesses:**

1) Although the authors mention the proposed tracker achieves 22.7 FPS on DanceTrack, the impact on computational speed from the proposed modules remains unclear. It's recommended to include speed comparison with the baseline method.
2) The proposed ReID feature transformation method is the major contribution of the paper. As far as I'm concerned, there are other works also making such efforts, e.g. TOPICTrack. Comparisons or discussions on such related work are encouraged, if possible.
3) Exploiting ReID cues is not new for MOT, this paper essentially seems incremental work, so the noveltiy is somewhat limited. If a well enough ReID model emerges, there is no need to optimize and improve it.
4) The performance imprvements are limited on MOT17. Additionally, What's about its results on MOT20?
5) The introduced extra hyper-parameters brings the difficulty for generalization.
6) All experiments are conducted for SORT-like methods, Can the proposed HAT benefical for E2E MOT?

**Questions:**

See Weaknesses

---

> ### Author Response · Authors · 2025-11-21
> **Response by Authors about Inference Speed**
>
> We sincerely thank the reviewer for the comment. Our replies addressing the reviewer's questions are presented below:
>
> 1. **FPS:**
>     - On the same hardware, the baseline achieves an efficiency of 46.5 FPS (without the detector). Although our method incurs an efficiency overhead, as discussed in L906-911, **the main reason is the lack of a GPU-friendly `eigh()` operator**. Based on existing evidence, **we anticipate that if this operator gains future GPU support, its computational efficiency could increase by 4 to 10 times**, as it currently represents the dominant computational cost in our entire algorithm.
>     - We believe that the implementation of this specific operator falls outside the scope of this paper. As an alternative, if our method is deployed in a practical application, one could consider performing an extra dimensionality reduction step on the features to alleviate the computational burden of the `eigh()` operator on the CPU. This strategy has proven effective in accelerating inference in our experiments.

---

> ### Author Response · Authors · 2025-11-21
> **Response by Authors about Related Work and Novelty**
>
> 2. **Related Work and Novelty:**
>     - The main objective of TOPICTrack [o], mentioned by the reviewer, is to propose a novel tracking scenario and its corresponding dataset (BEE24). For the method, they mainly demonstrate how to integrate motion and appearance tracking cues via a dual-branch algorithm. The handling of ReID only occupies a very small part of the paper and lacks sufficiently clear analysis and argumentation. Furthermore, based on our understanding, the transformation they apply to ReID features functions more like a polarization process rather than a true feature space transformation. As shown in their core code below, it essentially magnifies the difference between positive and negative scores, but does not alter the space where the similarity metric resides.
>         ```python
>         def _nn_res_recons_cosine_distance(x, y, tmp=100, data_is_normalized=False):
>             if not data_is_normalized:
>                 x = np.asarray(x) / np.linalg.norm(x, axis=1, keepdims=True)
>                 y = np.asarray(y) / np.linalg.norm(y, axis=1, keepdims=True)
>
>             ftrk = torch.from_numpy(np.asarray(x)).half().cuda()
>             fdet = torch.from_numpy(np.asarray(y)).half().cuda()
>             aff = torch.mm(ftrk, fdet.transpose(0, 1))
>             aff_td = F.softmax(tmp*aff, dim=1)
>             aff_dt = F.softmax(tmp*aff, dim=0).transpose(0, 1)
>
>             res_recons_ftrk = torch.mm(aff_td, fdet)
>             res_recons_fdet = torch.mm(aff_dt, ftrk)
>
>             sim = (torch.mm(ftrk, fdet.transpose(0, 1)) + torch.mm(res_recons_ftrk,
>                                                                 res_recons_fdet.transpose(0, 1))) / 2
>             distances = 1-sim
>
>             distances = distances.detach().cpu().numpy()
>             sim = sim.detach().cpu().numpy()
>
>             return distances, sim
>         ```
>     - Our paper, conversely, provides a **comprehensive investigation comprising an intuitive explanation of the sub-optimality of conventional ReID features, visualizations of specific cases, and precise global numerical analysis**. Following this, we proposed our method and subsequently validated its effectiveness and robustness using **tracking performance, specific case analyses, and global numerical results**. To our knowledge, this complete and comprehensive exploration and argumentation are absent in previous work, which is sufficient to represent the novelty of our contribution.
>     - The reviewer said *Exploiting ReID cues is not new for MOT, this paper essentially seems incremental work, so the novelty is somewhat limited*. We contend that this perspective appears somewhat metaphysical, and we would appreciate it if the reviewer could provide relevant references to substantiate this claim. We believe that while this potential idea may be (but we don't know) implied in some papers, our specific approach to "finding a better representation space constructed by historical trajectories", which is a core concept unique to MOT, has not been previously articulated and fully investigated.
>     - The reviewer said *If a well enough ReID model emerges, there is no need to optimize and improve it*. We present two aspects of evidence to alleviate the reviewer's apprehension. Firstly, as for **experimental results, both the classical FastReID (Tab.1 & Tab.2) and the latest MASA (Tab.3)** (which is well-trained on ultra-large datasets) still show room for improvement when augmented by our method. It is foreseeable that, at least for the next several years, our approach can elevate the performance ceiling of ReID models, simply because they are not yet perfect. Secondly, from the **perspective of domain evolution, perfection is our ultimate goal, but the contributions made in the interim before absolute perfection is reached are undeniably valuable**. For instance, although we may one day eliminate computational resource constraints, the current research on sparsity and efficient computation remains highly significant and indelible.
>
>
> [o]. Cao, Xiaoyan, et al. "TOPIC: a parallel association paradigm for multi-object tracking under complex motions and diverse scenes." IEEE Transactions on Image Processing (2025).

---

> ### Author Response · Authors · 2025-11-21
> **Response by Authors for Experiments**
>
> 3. **MOT20 Results:**
>     - Given that the scenes in MOT17 and MOT20 are nearly identical, with some video sequences even being shared, we anticipate that the results on these two benchmarks would not differ significantly. Our method is dedicated to improving the discriminability of ReID representations, but if a dataset's appearance similarity is too low, it will limit the performance gain our method can provide. We consider this to be reasonable and acceptable, as evidenced in Fig.4 of our paper. This is why we did not initially provide MOT20 results, **as we believed the MOT17 results are sufficiently representative**.
>     - Due to the submission limitations of the MOT challenge (i.e., only three submission opportunities per individual during our rebuttal phase) and the server instability, obtaining timely results is highly difficult. This, combined with the absence of a proper validation set, poses a non-trivial challenge for rapid experimentation. We are actively conducting these experiments and anticipate being able to share some preliminary findings.
> 4. **Hyper-parameters:**
>     - While our approach introduces three hyperparameters, our experimental results (presented in Tab.5, Tab.6, and Tab.7) indicate that a noticeable performance improvement over the baseline (#5 in Tab.4) is maintained, even with settings that substantially deviate from the optimal. This indicates that **our method can obtain stable performance gains even without hyperparameter tuning**. This demonstrates the robustness of our method to hyperparameter settings.
>     - We performed hyperparameter tuning to explore a better performance upper bound, a practice that is common across all current heuristic algorithms. Crucially, we ensured that the hyperparameters for our baseline and SOTA methods were consistent in our experiments, thereby **confirming that our observed improvement is not attributed to hyperparameter tuning**, thus guaranteeing validity.
> 5. **For End-to-End Methods:**
>     - Regarding this point, we recommend that the reviewer read Lines 987–998, as well as Point 3 of our [Response to Reviewer veDF](https://openreview.net/forum?id=G5YWhGslEr&noteId=oJua1H8AHX).
>     - In terms of direct technical integration, FLD changes the vector's dimensionality, preventing its immediate application to well-trained E2E models. This situation is both understandable and acceptable, given that many concepts and techniques originating in heuristic algorithms and E2E models have historically been challenging to integrate directly, as they stem from two distinct technological development paths.
>     - Beyond the specific implementation, the main contribution lies in offering inspirational insights that may hold great potential to motivate the design philosophy of next-generation E2E models. This could include, for instance, the necessity of explicitly designing feature interactions between different trajectories.
>
> We hope that all the responses provided above sufficiently address the reviewer's confusion. If there are any questions, we welcome continued discussion.

---

> ### Author Response · Authors · 2025-12-02
> **Response by Authors for MOT20 Results**
>
> We conducted comprehensive experiments using the FastReID-MOT as the baseline on MOT20. Unfortunately, due to the platform having changed in the last few months, the MOT20 evaluation system has some issues with evaluation failures. Please refer to the [official forum](https://www.codabench.org/forums/9893/1624/) for specific details (login required for viewing).
>
> Given the issues mentioned above, we can only rely on the MOT20-val results for this rebuttal, which are presented as follows:
> | Method           | HOTA | DetA | AssA | MOTA | IDF1 |
> |------------------|------|------|------|------|------|
> | Deep OC-SORT     | 59.5 | /    | 58.2 | /    | 76.3 |
> | Hybrid-SORT-ReID | 60.7 | 61.6 | 60.0 | 74.0 | 78.4 |
> | FastReID-MOT (baseline)     | 57.7 | 61.7 | 54.1 | 74.5 | 72.4 |
> | HAT-FastReID-MOT (ours) | **61.2** | **62.3** | **60.4** | **75.0** | **78.8** |
>
> All the aforementioned methods utilize the same publicly available FastReID weight, ensuring a fair comparison.
>
> Our proposed method demonstrates significant improvements over both advanced algorithms (Hybrid-SORT/Deep OC-SORT) and our baseline. This verifies the feasibility of our approach on MOT20 and further validates its generalization and robustness across diverse scenarios.

---

### Official Review · Reviewer_veDF · 2025-10-30

**Soundness:** 2
**Presentation:** 2
**Contribution:** 2
**Rating:** 4
**Confidence:** 4

**Summary:**

this paper investigates the visual similarity measurements for multi-object tracking (MOT). specifically, the authors argue that the default approach of re-ID feature distance might be sub-optimal for the tracking task, since re-ID aims to distinguish all potential targets, whereas tracking aims to differentiate similar targets within the same video sequence. as a solution, the paper treats the historical trajectory features as conditions and adopts Fisher Linear Discriminant (FLD) to project the original re-ID feature into a different feature space. on multiple benchmarks, the proposed feature achieves improvements over the baseline.

**Strengths:**

+ the observation that the re-ID feature might be sub-optimal for tracking is very interesting.
+ the feature dimension reduction method is easy to implement and introduces an additional 'degree-of-freedom' for test-time optimization to the feature-distance-based visual similarity calculation.
+ consistent improvements over the baseline across multiple benchmarks.

**Weaknesses:**

- no new insights & high similarity to undiscussed existing work [r1]. [r1] also points out that re-ID feature might not be the optimal choice for tracking, and conducted experiments to verify the mismatch. given the specific points listed below, it is very concerning that this paper does not compare against it.
  - motivation - reID feature & matching scope in tracking: Fig. 3 in [r1] vs. Fig. 1 in this paper
  - issue in existing methods - similarity estimation & mismatch between reID and tracking: section III in [r1] vs. section 2.1-2.2 in this paper
  - preliminary verification - matching performance improvements from the proposed metric: Fig. 4 in [r1] vs. Fig. 2b in this paper
- ablation & variant study.
  - in Table 5, the main ablation for the proposed dimension reduction with FLD should be a variant where no feature dimension reduction is conducted. is it the $d=d'$ variant? are variants #1 and #5 the same as the baseline? or do they still include the Temporal-Shifted Trajectory Centroid?
  - in Table 6, the main ablation should be the setting where no Temporal-Shifted Trajectory Centroid is incorporated, not the baseline tracker. is it the $T=inf$ line?
  - in Table 7, the main ablation should be the version where only the traditional re-ID distance (NO projection) is considered, which should be $\alpha=0$. the reviewer cannot find this variant.
- this method is not applicable to end-to-end tracking approaches
- no comparison with SOTA methods such as MOTRv2 (CVPR 2023) and ColTrack (ICCV 2023)

[R1]. Hou, Yunzhong, Zhongdao Wang, Shengjin Wang, and Liang Zheng. "Adaptive affinity for associations in multi-target multi-camera tracking." IEEE Transactions on Image Processing 31 (2021): 612-622.

**Questions:**

see above

---

> ### Author Response · Authors · 2025-11-18
> **Response by Authors for Weakness 1 and 2**
>
> We sincerely appreciate your detailed comments and suggestions. I apologize for the slight delay in this response, which was unfortunately due to some personal health reasons.
>
> We are highly grateful for and fully agree with the reviewer's positive assessment of our contribution. The central goal of our study is clearly summarized: **to demonstrate the limitations of conventional ReID approaches within the MOT framework, and consequently, to provide an accessible and direct baseline that motivates the development of more sophisticated and tailored methodologies.**
>
> Our detailed response to the reviewer's concerns is as follows:
> 1. **Related Work Discussion:**
>     - Due to the page limit, we have included a brief discussion of the paper [a] mentioned by the reviewer in our appendix (L711-712).
>     - In detail, while both papers explore ReID features, **our motivation and core contributions diverge significantly**. The authors in [a] argue that performing global temporal ReID feature matching is sub-optimal for tracking. Although their approach involves adjusting ReID features, **their core focus lies in the** ***temporal dimension***, utilizing an additionally trained model to perform local temporal refinement for a single input feature. In contrast, **our method centers on how to construct a superior subspace for tracking by modeling the relationships** ***between different historical trajectories***. This markedly differentiates the fundamental idea and motivation of our work from that of [a].
>     - Fig.3 in [a] is intended to contrast the differences between temporal global and local behaviors, primarily serving as an illustration of metric calculation methods. In contrast, our Fig.1 presents a data analysis sampled from real data, which aims to reveal the feature distribution disparities between different trajectories and videos. The focus of these two figures is divergent.
>     - Section III.C and Fig.4 in [a] also exclusively discuss the difference between global and local matching in the temporal dimension.
> 2. **Ablation Issues:**
>     - Yes, the $d' = d$ is the variant where no feature dimension reduction is conducted. The baselines #1 and #5 do not apply any techniques we proposed in this paper, including Temporal-Shifted Trajectory Centroid.
>     - In Tab.5, all experiments are conducted without the use of Temporal-Shifted Trajectory Centroid. As stated in L430-431, our ablations proceed incrementally, and the role of Temporal-Shifted Trajectory Centroid is introduced and explored starting from Tab.6. Therefore, all experiments presented in Tab.5 are equivalent to using naive averaging with $\lambda_0=1.0$.
>     - Since all techniques ablated in Tab.5 and Tab.6 are applied exclusively to the projected feature branch, the result presented in Tab.7 with $\alpha=0.0$ is precisely equivalent to the baseline #5 performance shown in Tab.4.
>
>
> [a]. Hou, Yunzhong, Zhongdao Wang, Shengjin Wang, and Liang Zheng. "Adaptive affinity for associations in multi-target multi-camera tracking." IEEE Transactions on Image Processing 31 (2021): 612-622.

---

> ### Author Response · Authors · 2025-11-18
> **Response by Authors for Weakness 3 and 4**
>
> 3. **Apply on E2E methods:**
>
>     - We refer the reviewer to the Appendix (L987-998) for a comprehensive discussion on this matter.
>
>     - Overall, we contend that end-to-end and heuristic algorithms represent two distinct developmental paths. Due to the significant differences in problem formulation and algorithmic design, a seamless and perfect transfer of techniques between them is indeed infeasible. However, **we argue that this should not be viewed as a fatal flaw**—akin to how the specialized solutions from Classical Mechanics are not directly interchangeable with those of Quantum Physics, yet both remain crucial for their respective domains.
>
>     - Beyond the specific method, **the more critical contribution of our paper is identifying a previously overlooked problem and providing inspirational insights**. We firmly believe this line of thinking holds great potential to motivate the design of future E2E algorithms. For example, if one aims to design an E2E feature matching model, the discussions presented in our work suggest the need to fully consider and design feature interaction between different trajectories to enhance the representation capacity for a specific video sequence.
>
> 4. **Comparisons with E2E methods:**
>
>     - Due to page constraints and the observation that E2E and heuristic methods exhibit drastically different dominance across various datasets (e.g., E2E methods demonstrate a significant lead over all heuristic algorithms on DanceTrack), we opt to exclude these comparison results from the submitted version. We posit that **benchmarking against E2E methods lacks fair evaluation criteria due to the fundamental difference in their operational paradigms**. For example, as seen in the comparison between the DETR and YOLO series, varying formulations and architectures lead to differing strengths and weaknesses depending on the specific application scenario. However, it does not diminish the fact that both paradigms remain equally crucial and warrant continued development. We will articulate and discuss this perspective in the subsequent version of the manuscript.
>
>     - We agree with the reviewer that this discussion is crucial to prevent potential confusion for the reader, and **we will incorporate it into the final version** (including MOTRv2 and ColTrack). On DanceTrack, our method is competitive with the newly published CO-MOT [b] and SambaMOTR [c], i.e.,  66.9 (ours), 65.3 (CO-MOT), and 67.2 (SambaMOTR). On SportsMOT, our method surpasses all existing published end-to-end methods (for example, 78.9 (ours) *vs.* 72.6 (MOTIP [d], CVPR 2025)) with a large margin.
>
>
>
> We sincerely hope that the responses above adequately address the reviewer's concerns. Should there be any further questions or points requiring clarification, we welcome the continued and deep discussion.
>
>
>
> [b]. Luo, Weixin, et al. "CO-MOT: Boosting End-to-end Transformer-based Multi-Object Tracking via Coopetition Label Assignment and Shadow Sets." The Thirteenth International Conference on Learning Representations. 2025 (ICLR 2025).
>
>
>
> [c]. Mattia Segu, et al. "Samba: Synchronized Set-of-Sequences Modeling for Multiple Object Tracking." The Thirteenth International Conference on Learning Representations. 2025 (ICLR 2025).
>
>
>
> [d]. Gao, Ruopeng, Ji Qi, and Limin Wang. "Multiple object tracking as id prediction." Proceedings of the Computer Vision and Pattern Recognition Conference. 2025 (CVPR 2025).

---

### Official Review · Reviewer_ADMy · 2025-10-30

**Soundness:** 3
**Presentation:** 2
**Contribution:** 2
**Rating:** 2
**Confidence:** 3

**Summary:**

This paper proposes a history-aware feature transformation method that dynamically creates more discriminative subspaces using Fisher Linear Discrimination (FLD), tailored to each video's unique sample distribution.
The proposed method treats the history features of established trajectories as context and projects adjusted raw ReID features into a video-specific representation space.
The effectiveness of the proposed method is discussed by comparing it with several tracking methods using multiple datasets.
The paper states that the code will be made publicly available.

**Strengths:**

- The paper is easy to understand.
- The proposed method is a combination of elementary techniques, making it straightforward to comprehend.

**Weaknesses:**

- Insufficient experiments
  - Generally, tracking methods are comprehensively compared using multiple metrics like IDF1 and MOTA.
  - Specifically, these include IDF1, IDP, IDR, Recall, Precision, FP, FN, IDs, FM, MOTA, IDt, IDa, IDm, etc. This paper evaluates only a very limited subset of these metrics.

  - Furthermore, comparisons with transformer-based methods like MOTR, MOTRv2, and their variants are absent. It remains unclear whether the proposed method outperforms approaches like MOTRv2, especially on datasets such as DanceTrack.

  - Furthermore, the proposed method is not compared against more general methods like MOT20 or MOT17.
  - Consequently, it is difficult to assess the effectiveness of the proposed method.

- While the proposed method states it “requires no training,” FLD itself is a form of machine learning. Moreover, the proposed method is a combination of rudimentary techniques, lacking technical depth.

**Questions:**

- Why was it not compared against more general methods like MOTA or IDF1?
- Why was it not compared against more sophisticated transformer-based methods like MOTR or MOTRv2?
- Why was it not compared against more general methods like MOT20 or MOT17?

---

> ### Author Response · Authors · 2025-11-20
> **Response by Authors for More Metrics**
>
> We appreciate the reviewer's feedback. We note the reviewer's reduction of the confidence score on the comment (from 3 to 1), which suggests that there may be some misunderstandings requiring clarification, perhaps stemming from the reviewer not specializing in the multi-object tracking (MOT) domain.
>
> 1. **More Metrics:**
>     - The foremost reason is that, due to space constraints, it is difficult for us to include all possible MOT evaluation metrics.
>     - The more fundamental reason is that metrics such as **MOTA and IDF1 have been superseded by the more comprehensive metric, HOTA, in recent years**. For example, MOTA inherently has a strong bias toward detection performance, consequently neglecting the critical importance of trajectory consistency and coherence. **This perspective is frequently discussed and well-established in recent top-conference publications [b, d, e, f, g], thus reflecting the current consensus of the MOT community**. This is also why, if the reviewer examines **recently published SOTA methods, they will find that their MOTA/IDF1 metrics do not necessarily show continuous improvement, as researchers are prioritizing HOTA-series metrics**. Explicit notification can be found in [b], where they also emphasize the use of HOTA as the primary metric in both Tab.2 and Tab.3.
>     - Certainly, if our revised manuscript allows for sufficient space, we would be able to include the results for these metrics. However, we believe they are not crucial, which is why we opted to omit them when faced with space limitations.
>     - To mitigate the reviewer's confusion, we can provide the results of these two metrics for our main results on DanceTrack here:
>         - HAT-FastReID-MOT$\dagger$: 89.7 MOTA, 61.1 IDF1.
>         - HAT-OC-SORT-ReID: 90.3 MOTA, 67.7 IDF1.
>     - As for the remaining metrics mentioned by the reviewer—IDP, IDR, Recall, Precision, FP, FN, IDs, FM, IDt, IDa, and IDm—they have not been referenced in any top-conference papers [b, c, d, e, f, g, h, i, j, k] published in recent years, as far as we know. This is because they are too detailed, lacking justifiable meaning and representative significance.
>
> [b]. Luo, Weixin, et al. "CO-MOT: Boosting End-to-end Transformer-based Multi-Object Tracking via Coopetition Label Assignment and Shadow Sets." The Thirteenth International Conference on Learning Representations. 2025 (ICLR 2025).
>
> [c]. Mattia Segu, et al. "Samba: Synchronized Set-of-Sequences Modeling for Multiple Object Tracking." The Thirteenth International Conference on Learning Representations. 2025 (ICLR 2025).
>
> [d]. Gao, Ruopeng, Ji Qi, and Limin Wang. "Multiple object tracking as id prediction." Proceedings of the Computer Vision and Pattern Recognition Conference. 2025 (CVPR 2025).
>
> [e]. Zeng, Fangao, et al. "Motr: End-to-end multiple-object tracking with transformer." European conference on computer vision. Cham: Springer Nature Switzerland, 2022 (ECCV 2022).
>
> [f]. Gao, Ruopeng, and Limin Wang. "MeMOTR: Long-term memory-augmented transformer for multi-object tracking." Proceedings of the IEEE/CVF International Conference on Computer Vision. 2023 (ICCV 2023).
>
> [g]. Luiten, Jonathon, et al. "Hota: A higher order metric for evaluating multi-object tracking." International journal of computer vision 129.2 (IJCV 2021): 548-578.
>
> [h]. Qin, Zheng, et al. "Towards generalizable multi-object tracking." Proceedings of the IEEE/CVF Conference on Computer Vision and Pattern Recognition. 2024 (CVPR 2024).
>
> [i]. Mancusi, Gianluca, et al. "Is multiple object tracking a matter of specialization?." Advances in Neural Information Processing Systems 37: 133776-133800 (NeurIPS 2024).
>
> [j]. Liu, Yiheng, Junta Wu, and Yi Fu. "Collaborative tracking learning for frame-rate-insensitive multi-object tracking." Proceedings of the IEEE/CVF international conference on computer vision. 2023 (ICCV 2023).
>
> [k]. Mancusi, Gianluca, et al. "Trackflow: Multi-object tracking with normalizing flows." Proceedings of the IEEE/CVF International Conference on Computer Vision. 2023 (ICCV 2023).
>
> [l]. Seidenschwarz, Jenny, et al. "Simple cues lead to a strong multi-object tracker." Proceedings of the IEEE/CVF conference on computer vision and pattern recognition. 2023 (CVPR 2023).

---

> ### Author Response · Authors · 2025-11-20
> **Response by Authors for More Comparisons**
>
> 2. **Compare with End-to-End Methods:**
>     - All missing competitors mentioned by the reviewer are end-to-end (E2E) MOT models. For a detailed explanation and statement regarding this point, we suggest the reviewer refer to our **Appendix (L987–998)** and our **[response to the reviewer veDF](https://openreview.net/forum?id=G5YWhGslEr&noteId=oJua1H8AHX), point 4**.
>     - In summary, E2E and heuristic algorithms lack fair and equivalent comparison conditions, exhibiting mixed performance across different datasets, thus rendering direct comparison less meaningful. As for the results, our method is competitive with newly published E2E approaches [b, c] on DanceTrack and significantly surpasses the existing E2E SOTA [d] with a large margin on datasets like SportsMOT.
>     - The reason we omitted this comparison is the page limit, which is why we focused more extensively on internal comparisons among heuristic algorithms. **This is also a common convention observed in some currently published literature [k, m] focusing on heuristic algorithms**.
>
> 3. **Pedestrian Tracking Results:**
>     - Regarding the general pedestrian tracking mentioned by the reviewer, **we present results on MOT17 and provide detailed analysis and argumentation in Appendix C.1**.
>     - Overall, our method still achieves a stable improvement on MOT17. However, in the existing pedestrian tracking datasets, the visual distinctiveness of pedestrians is already quite clear (see Fig.4), which prevents our method's advantages from being demonstrated more prominently. We consider this outcome to be both intuitively reasonable and acceptable.
>
> [b]. Luo, Weixin, et al. "CO-MOT: Boosting End-to-end Transformer-based Multi-Object Tracking via Coopetition Label Assignment and Shadow Sets." The Thirteenth International Conference on Learning Representations. 2025 (ICLR 2025).
>
> [c]. Mattia Segu, et al. "Samba: Synchronized Set-of-Sequences Modeling for Multiple Object Tracking." The Thirteenth International Conference on Learning Representations. 2025 (ICLR 2025).
>
> [d]. Gao, Ruopeng, Ji Qi, and Limin Wang. "Multiple object tracking as id prediction." Proceedings of the Computer Vision and Pattern Recognition Conference. 2025 (CVPR 2025).
>
> [k]. Mancusi, Gianluca, et al. "Trackflow: Multi-object tracking with normalizing flows." Proceedings of the IEEE/CVF International Conference on Computer Vision. 2023 (ICCV 2023).
>
> [m]. Yi, Kefu, et al. "Ucmctrack: Multi-object tracking with uniform camera motion compensation." Proceedings of the AAAI conference on artificial intelligence. Vol. 38. No. 7. (AAAI 2024).

---

> ### Author Response · Authors · 2025-11-20
> **Response by Authors for "Training-Free" Claim**
>
> 4. **Training-Free Claim:**
>     - There appears to be a discrepancy in the definition of *training-free* between us and the reviewer. We maintain that since **our method does not consume any training data** and **lacks any modules involving gradient backpropagation**, it should not be considered training-required by the current research community.
>     - Although FLD is considered a machine learning technique, this does not imply it is training-based. We only need to compute the **closed-form solution** based on historical trajectories (inference-time data). We believe that in the current deep learning era, this **should not be classified as a training outcome, but rather as a form of numerical analysis**.
>     - The most convincing evidence is that the ICLR-published article [n] employed a Gaussian Discriminant Analysis (GDA) method, similar to FLD, as its core algorithm, and asserts that it is training-free. **This clearly demonstrates that our claim is accepted and acknowledged by the top conference.**
>
>
> We hope that the three responses provided above will help eliminate the misunderstandings and address the reviewer's concerns. If any other issues remain, we would be happy to discuss them further.
>
> [n]. Wang, Zhengbo, et al. "A Hard-to-Beat Baseline for Training-free CLIP-based Adaptation." The Twelfth International Conference on Learning Representations (ICLR 2024).

---

> ### Comment · Reviewer_ADMy · 2025-11-20
> **Comment for  Authors**
>
> First and foremost, thank you very much for your sincere feedback.
> I also deeply apologize for the fact that someone like me, a non-specialist in pattern recognition and computer vision reviewed your paper.
> I am also grateful for your sincere comments regarding this review result. (A reviewer provided meaningful suggestions regarding my very poor expertise, which made me reconsider my own significant poor expertise in computer vision and pattern recognition, fortunately.)
> Furthermore, I sincerely apologize that, during the majority of the review period (five papers in two weeks), I faced unexpected health issues, which  slightly reduced the time available for reviewing compared to normal.
> Considering these points, I downgraded my confidence rating from 3 to 1.
> Now, I would like to comment on the sincere feedback you provided.
>
> ---
>
> #### Regarding Evaluation Metrics
> As you pointed out, I am aware (even as a non-specialist) that HOTA is widely used as a major metric in academic papers. I generally agree with the authors' perspective.
>
> By presenting multiple metrics in the supplementary materials, I believed I could gain a more detailed understanding of the strengths and weaknesses of the proposed method. Based on my layperson's understanding, I had assumed these could be easily computed using the publicly available code [1].
>
> [1] https://github.com/cheind/py-motmetrics
>
> In my non-expert understanding, metrics like MOTA, HOTA, and IDF1 are ultimately comprehensive metrics that take multiple metrics as arguments.
> Therefore, even if the overall value (e.g., MOTA) is high, it is not uncommon for some of its components (e.g., the arguments of MOTA) to be significantly inferior.
> Thus, I believed that knowing the detailed values with various metrics would clarify the limitations and make the discussion between the reviewer and the author more constructive.
>
> In my non-expert understanding, ultimately, which metric is most important depends on the use case in actual scenarios. For instance, in actual scenarios requiring continuous tracking even in heavily occluded environments, IDs might be more critical than HOTA.
> That's why I thought it would be helpful to include the detailed metrics in the supplementary materials.
>
> However, I agree with the authors' position, so further discussion is unnecessary.
> If you would like to pursue further discussion, please do not hesitate to let me know.
>
>
> ---
>
> #### Comparison with E2E and Pedestrian Tracking Results
> First, thank you for the thorough discussion. I appreciate it.
>
>
> Also, I had overlooked the fact that the MOT17 results were included in the supplementary materials. I deeply apologize for this oversight.
>
>
> First, I agree with the author's perspective on E2E.
> Now, please note that my point of discussion here was not the naive and childish viewpoint that “since the proposed method performs worse than the YYY-method on the XXX-dataset, it is inferior to conventional methods.”
>
>
> My concern was simply the fact that a comparison with E2E was not included.
> Even if the proposed method was numerically inferior to the SOTA E2E method, I thought that presenting those results and analyzing the issues would lead to a meaningful discussion for reviewers, including myself.
>
>
> I am not an expert in computer vision and pattern recognition, but I believe that in general (including in MOT), the E2E approach is data-hungry and therefore tends to perform well when there is a large amount of data (e.g., dance tracks).
>
>
> On the other hand, since the proposed method is Re-ID-based, it is less data-hungry than the E2E approach.
> Therefore, I understand that the proposed method is competitive than the E2E-based method in cases such as MOT17 and MOT20. In my layman's understanding, MOT20 in particular is a benchmark introduced to focus on far more congested, challenging scenes than MOT17.
>
>
> For this reason, I highlighted the lack of experiments for MOT20, believing that the results of MOT20 would further strengthen the superiority of the proposed method.
>
>
> > However, it does not diminish the fact that both paradigms remain equally crucial and warrant continued development.
>
> I agree with the above point. Ultimately, the advantageous method and paradigm change depending on the target data, so both paradigms need to be continuously developed. Ultimately, I believe MOT is still very much in its developmental stage. This is likely because tracking requires more complex annotation compared to detection, identification, and action recognition.
>
>
> However, I also agree with the author's opinion, so further discussion is unnecessary.
> If you would like to discuss this further, please do not hesitate to let me know.

---

> ### Comment · Reviewer_ADMy · 2025-11-20
> **Comment for Authors**
>
> #### Training-Free Claim
>
> Thank you very much for your important and insightful discussion.
> I recognize that further discussion on whether an algorithm is “training-free” or not is not particularly beneficial.
> I believe it is necessary to summarize this discussion in a footnote or similar location.
> While I am not an expert in computer vision and pattern recognition, I understand that citing reference [n] as a fact is highly significant.
> Since you have provided the facts, I believe it is acceptable to prioritize the authors' “definition” of “training-free.”
>
> I agree with the authors' position, so further discussion is unnecessary. If you would like to pursue further discussion, please do not hesitate to let me know.
>
> ---
>
> However, given that discussion time is limited, I believe it would be beneficial to allocate time to discussions with other reviewers.
> Thank you very much.

---

> > ### Author Response · Authors · 2025-11-25
> > **Response by Authors**
> >
> > **We are delighted to have achieved consensus.**
> >
> > We will provide a wider range of metrics in the revised Supplementary Material. However, due to the lack of corresponding metrics in previously published works and the inherent reproduction errors, we can only guarantee the provision of complete metrics for our own method.
> >
> > Thank you for your understanding regarding our decision not to compare with end-to-end methods. If space permits, we will include the performance of some representative E2E models in the main text. Otherwise, we will discuss them in our Appendix.
> >
> > Regarding MOT20, we are currently experiencing issues with result submission. Since they adopted a new platform unfamiliar to us, we are still communicating with the organizers to resolve the matter. Under the standard half-validation set evaluation, **our method demonstrates an improvement of approximately 4.0 HOTA points over the baseline**. This further highlights the effectiveness of our approach.
> >
> > We are very grateful for the constructive discussion with the reviewer. **We are pleased to confirm that all prior misunderstandings have been eliminated, and we believe a consensus on the key points has been achieved. Given that, we respectfully request the reviewer to reconsider and adjust his/her rating based on the currently available information.**

---

> ### Comment · Reviewer_ADMy · 2025-11-26
> **Comment for Authors**
>
> Dear Author,
>
> Thank you very much for your thoughtful response.
>
> I also sincerely appreciate you sharing the preliminary results of the MOT20 experiment.
> I have raised my score by one level.
>
> On the other hand, for two reasons as explained above, I have changed the confidence level to 1.
> Therefore, I hope you understand that my score itself is highly unlikely to influence the decision on whether to accept or reject the paper. However, your insightful and expert comments will serve as invaluable reference material for the AC when making their decision.
>
> Finally, it has been an honor to learn so much from researchers and reviewers of such high expertise as yourself.

---

> > ### Author Response · Authors · 2025-11-27
> > **Response by Authors**
> >
> > Thank you very much for your prompt reply and constructive discussion. We are also deeply grateful for your decision to raise the rating for our work.
> >
> > Regardless of the outcome, your active and open-minded engagement preserves and enhances the spirit of openness and communication within the ICLR community.
> >
> > Additionally, we will compile and report the definitive results of our method on the MOT20 benchmark within the next day or two. We hope this further illustrates the effectiveness and robustness of our approach.

---

### Author Response · Authors · 2025-12-03
**Rebuttal Summary by Authors**

To facilitate the final decision by ACs and SACs, we summarize all key discussions from the rebuttal phase as follows:

1. Regarding the concerns raised by Reviewer ADMy, we ensured all questions were addressed through discussion. We also clarified several misunderstandings, as the reviewer is not an MOT expert. Accordingly, the reviewer agreed to revise his/her score to positive.

2. We also provided a detailed reply to the weaknesses raised by Reviewer veDF.
    1. The discussion concerning the related work was already included in the appendix of our initial submission. In [our response](https://openreview.net/forum?id=G5YWhGslEr&noteId=MhmxZEVcXN), we further detailed the differences, demonstrating that the distinctions between the two works are fundamental. This ensures the novelty of our work.
    2. We clearly detailed the procedure for the incremental ablation study in the main text of our initial submission (Section B.5) to prevent any reader misunderstanding. We have also fully addressed the reviewer's concerns in [our response](https://openreview.net/forum?id=G5YWhGslEr&noteId=MhmxZEVcXN).
    3. Regarding the comparison and discussion with E2E (End-to-End) models, we provided a detailed discussion in Appendix Section D of our initial submission, and further elaborated in our rebuttal response ([1](https://openreview.net/forum?id=G5YWhGslEr&noteId=oJua1H8AHX)/[2](https://openreview.net/forum?id=G5YWhGslEr&noteId=kdHkjlHldY)). It is a widely accepted consensus in MOT community that heuristic-based algorithms cannot be directly applied to E2E models, while a direct performance comparison is also not representative.

3. We provided a compelling rebuttal addressing the weaknesses raised by Reviewer zen2 in detail:
    1. We included the baseline results for inference speed in our [reply](https://openreview.net/forum?id=G5YWhGslEr&noteId=c1rq3NKyGz).
    2. Regarding the similar work, we provided code evidence and a detailed discussion in [our response](https://openreview.net/forum?id=G5YWhGslEr&noteId=z1aikpgfmd). This demonstrates that our work is fundamentally different from that paper, considering both the motivation and the implementation. This affirms the novelty of our approach. Furthermore, we also demonstrated the general applicability of our method.
    3. We also supplemented the MOT20 results requested by the reviewer in [our response](https://openreview.net/forum?id=G5YWhGslEr&noteId=FqwxScKfWJ).
    4. We also provided detailed explanations for the reviewer's other points of concern in [our response](https://openreview.net/forum?id=G5YWhGslEr&noteId=kdHkjlHldY). These points are consensus in the field of heuristic algorithms for MOT and are therefore acceptable.


Overall, we have provided reasonable explanations to address the concerns of all reviewers. The majority of the issues raised primarily requested additional experimental results or clarifications, including the discussion of similar work, all of which have now been completed. Crucially, no reviewer raised fundamental objections to the motivation or overall methodology of our paper.
All necessary revisions are included in the updated submission: MOT20 in Tab.9, the comparison with E2E models in Tab.11 and Tab.12, the detailed comparison with more metrics in Tab.10, and related work and discussions are included in Section A.

Finally, we would like to emphasize the highlights of this paper:

1. Our **core motivation is straightforward and natural**.
2. Based on this, we provide **a comprehensive investigation comprising an intuitive explanation of the sub-optimality of conventional ReID features, visualizations of specific cases, and precise global numerical analysis**. This offers sufficient and compelling evidence for our intuition.
3. Following this, we proposed our method and subsequently validated its effectiveness and robustness using **tracking performance, specific case analyses, and global analysis results**.

To our knowledge, ***this complete and comprehensive exploration of this sub-optimality is absent in previous work, which is sufficient to represent the novelty of our contribution***.

---

### Meta-Review · Area_Chair_8JRT · 2026-01-10

**Summary:**

The paper proposes a history-aware feature transformation method using Fisher Linear Discriminant to create more discriminative  ReID features for MOT.  However, significant concerns were raised：（1）Multiple reviewers highlighted a high similarity to prior work, questioning the paper's core novelty. （2）The ablation study was criticized as incomplete and comparisons with state-of-the-art transformer-based methods were absent.  As the authors' rebuttal failed to overcome the fundamental objections regarding the paper's novelty and experimental completeness, the AC recommends rejecting the paper.

The authors lodged a serious complaint that Reviewer veDF copied a review from a prior submission. While this is a serious matter for the program chairs to investigate independently, it does not negate the substantive technical criticisms raised, which are consistent with concerns from other reviewers.

**Reviewer Concerns:**

Addressed:

The authors provided a point-by-point response and included related work in the appendix.

The authors provided inference speed comparisons and a detailed discussion distinguishing their work from TOPICTrack.

Outstanding:

The rebuttal did not convincingly demonstrate fundamental novelty over [R1], as both works critically examine ReID features for MOT and propose adaptive refinements.

**Reviewer Scores:**

Reviewer ADMy: Likely raised from 2 to 3 , but with low confidence due to persistent issues.

Reviewer veDF: Unlikely to increase from 4 as core concerns (novelty, ablation, comparisons) remain unresolved.

Reviewer zen2: Expected to remain at 4. Provided analyses addressed minor points, but fundamental concerns about incremental contribution persist.

---

> ### Public Comment · ~Ruopeng_Gao1 · 2026-04-02
> **Factually Incorrectness about Meta-Review**
>
> I am the first author of this article. I am writing to place on record the unfair treatment we received during this public review process, specifically regarding the blatant factual errors and bias in the Area Chair's (AC) meta-review.
>
> Regarding Reviewer ADMy, the trajectory of their evaluation was as follows: Before we even submitted the rebuttal, the reviewer increased their score from 2 to 4 while lowering their confidence from 3 to 1, citing his/her own professional limitations. In the subsequent discussion (which took place before any alleged *information leak*), **Reviewer ADMy explicitly agreed to raise the score again—from 4 to 6**—and modified the final score accordingly. It is a documented fact that the reviewer had reached a positive consensus during our interaction.
>
> The AC, however, completely ignored this reality in the meta-review:
>
> - **Logical Inconsistency**: If the AC chose to base the decision on the *rolled-back* scores, Reviewer ADMy’s confidence should have been recorded as 3, not the 1 claimed by the AC.
> - **Disregard for Actual Consensus**: If the AC intended to reflect the changes made during the discussion, the final score should have been 6, not 4.
> - **Factual Fabrication**: The final score 3 guessed by the AC is a non-existent value that simply does not exist in the system for this reviewer.
>
> **In conclusion, the AC’s summary of our discussion with Reviewer ADMy is marred by significant factual errors and clear personal bias.**

---

> ### Public Comment · ~Ruopeng_Gao1 · 2026-04-02
> **Unjustified Reliance on Irresponsible Reviewer Comments**
>
> In our explicit comments to the ACs and PCs, I clearly stated that Reviewer veDF's comments were a verbatim copy-paste of his/her previous review for our manuscript submitted to ICCV 2025. However, the AC’s decision completely ignored the following critical points:
> - **Prior Resolution**: During the previous rebuttal process (ICCV 2025), Reviewer veDF explicitly claimed to have understood these points and subsequently raised their score to a positive rating.
> - **Irrelevance to the Current Version**: We formally notified the ACs and PCs that the **issues raised by veDF do not even exist in our current manuscript**. To proactively address these concerns, we had **even added a dedicated section (Section B.5)** for detailed discussion.
> - **Indisputable Misconduct**: The act of directly copying and pasting a previous review is evidenced by incontrovertible proof, establishing a clear case of unprofessional and irresponsible reviewing.
>
> Given the above facts, there is absolutely no logical or procedural justification for the AC to claim that *it does not negate the substantive technical criticisms raised*. Such a statement is not only a blatant disregard for the evidence provided but also a profound failure of responsibility.
>
> ***How can the AC justify validating criticisms that have been proven to be both obsolete and factually inapplicable to the current work?***

---

> ### Public Comment · ~Ruopeng_Gao1 · 2026-04-02
> **Copy of the Formal Report Regarding Irresponsible Reviewing Submitted to ACs and PCs**
>
> *For the sake of transparency, since our comments to the ACs and PCs are hidden from public view, I am reproducing a copy of the formal complaint we originally submitted below.*
>
> Dear Area Chairs and Program Chairs,
>
> Due to the recent AC re-assignment, we are reposting this urgent complaint to ensure you are aware of **a severe case of academic misconduct by Reviewer veDF**.
>
> We have irrefutable evidence that Reviewer veDF simply copied and pasted their previous comments from our ICCV 2025 submission without reading the current ICLR 2026 manuscript.
>
> 1. **Identical Text:** The review is verbatim identical to the old one.
> 2. **The Same Wrong References:** The reviewer incorrectly references the table numbers for our ablation studies (e.g., citing Table 5 instead of Table 4). This error is identical to the one in his/her previous review.
> 3. **Ignoring Facts:** All raised issues were already resolved in the previous revision, where this same reviewer previously acknowledged the fixes and raised their score.
>
> **A detailed comparison of the two reviews is provided below:**
>
> ICCV 2025:
> > * in Table 5, the main 'ablation' for the proposed dimension reduction with FLD should be a variant where no feature dimension reduction is conducted. is it the $D' = D$ indicates vairant? are variant #1 and #5 the same as the baseline? or do they still include the Temporal-Shifted Trajectory Centroid?
> > * in Table 6, the main 'ablation' should be a baseline where no Temporal-Shifted Trajectory Centroid is incorporated. is it the $T = \text{inf}$ line?
> > * in Table 7, the main 'ablation' should be the version where only the traditional re-ID distance (NO projection) is considered, which should be $\alpha = 0$. the reviewer cannot find this variant.
>
> ICLR 2026 (this time):
> > * in Table 5, the main ablation for the proposed dimension reduction with FLD should be a variant where no feature dimension reduction is conducted. is it the $d' = d$ variant? are variants #1 and #5 the same as the baseline? or do they still include the Temporal-Shifted Trajectory Centroid?
> > * in Table 6, the main ablation should be the setting where no Temporal-Shifted Trajectory Centroid is incorporated, not the baseline tracker. is it the $T = \text{inf}$ line?
> > * in Table 7, the main ablation should be the version where only the traditional re-ID distance (NO projection) is considered, which should be $\alpha = 0$. the reviewer cannot find this variant.
>
> Further details can be found in [our previous comment](INSERT_LINK_HERE). Furthermore, regarding the comparison with related work, we have thoroughly addressed this in both our previous rebuttal and the current manuscript, explicitly clarifying that the relevance between the two works is marginal. **Crucially, this reviewer had already fully acknowledged our arguments during the ICCV 2025 review process and subsequently raised their score to a positive rating.**
>
> *In summary, this represents a gross violation of reviewing ethics. We respectfully request that you disregard this invalid review entirely and consider disciplinary action against such irresponsible behavior.*
>
> Please note that, as we cannot upload images in the comment section, we are unable to attach the screenshots regarding the ICCV 2025 review. If required, please instruct us on how to submit this concrete evidence.

---

> ### Public Comment · ~Ruopeng_Gao1 · 2026-04-02
> **Unfounded Negative Bias**
>
> In the meta-review, the AC justified the rejection decision based on three primary grounds:
>
> - The similarity to related works.
> - The perceived incompleteness of the ablation studies.
> - The lack of comparison with transformer-based methods.
>
> **However:**
>
> - We have exhaustively discussed this in both our rebuttal and the Related Work section. During the rebuttal phase, we even provided the source code of the allegedly similar work to irrefutably demonstrate that its underlying methodology is fundamentally different from ours.
> - As discussed above, this alleged issue simply does not exist.
> - During the rebuttal phase, we explicitly provided the comparison results against transformer-based methods and incorporated them into the revised manuscript (Tables 11 and 12). Furthermore, we provided a clear rationale for why we consider these results to be of secondary importance (which explains their absence in the initial submission).
>
>
> The meta-review simply regurgitates the reviewers' initial comments in a meaningless manner, demonstrating a complete lack of critical synthesis or in-depth reflection.
>
> ***In conclusion, relying on fabricated facts, ignored rebuttals, and blind copy-pasting is not just unprofessional—it is utterly absurd. Such a meta-review makes a mockery of the peer-review process, and we strongly protest this gross editorial negligence.***

---

> ### Public Comment · ~Ruopeng_Gao1 · 2026-04-02
> **Open to Further Discussion**
>
> In the spirit of transparency and academic integrity, we remain fully open to communication. We welcome anyone—including the ACs, PCs, or any researchers who are interested in or have questions about our work—to engage with us directly here. We are more than happy to address any genuine inquiries.

---

### Decision · Program_Chairs · 2026-01-26

Reject